# River ice and water velocities using the Planet optical cubesat constellation

Andreas Kääb[1], Bas Altena[1], Joseph Mascaro[2]

[1]Department of Geosciences, University of Oslo, Oslo, 0316, Norway
[2]Planet, San Francisco, postal code 94103, USA

*Correspondence to*: Andreas Kääb (kaeaeb@geo.uio.no)

**Abstract.** The PlanetScope constellation consists of ~150 optical cubesats that are evenly distributed like strings of pearls in two orbital planes, scanning the Earth's land surface once per day with ~3 m spatial image resolution. Subsequent cubesats in each of the orbital planes image the Earth surface with a nominal time lapse of ~90 s between each other, which produces near-simultaneous image pairs over the across-track overlaps of the cubesat swaths. We exploit this short time lapse between subsequent Planet cubesat images to track river ice floes on Northern rivers as indicators of water surface velocities. The method is demonstrated for a 60 km long reach of the Amur River in Siberia, and a 200 km long reach of the Yukon River, Alaska. The accuracy of the estimated horizontal surface velocities is on the order of $\pm 0.01$ m s$^{-1}$. The application of our approach is complicated by cloud cover and low sun angles at high latitudes during the periods where rivers typically carry ice floes, and by the fact that the near-simultaneous swath overlaps by design do not cover the complete Earth surface. Still, the approach enables direct remote sensing of river surface velocities at a number of locations of many cold-region rivers and occasionally several times per year – much more frequent and over much larger areas than feasible so far. We find that freeze-up conditions seem to offer ice floes that are in general more suitable for tracking, and over longer time periods, compared to typical ice break-up conditions. The coverage of river velocities obtained could be particularly useful in combination with satellite measurements of river area, and river surface height and slope.

## 1 Introduction

Knowledge about water-surface velocities on rivers supports understanding a wide range of processes. In cold regions, river-ice freeze-up, and in particular break-up, and the associated transport of and action by ice debris is often the most important hydrological event of the year, producing flood levels typically exceeding those for other periods (Fig. 1) and with dramatic consequences for river ecology and infrastructure (e.g., Prowse et al., 2007; Kääb and Prowse, 2011; Rokaya et al., 2018a). River discharge measurements are complicated during freeze-up and break-up due to the physical impact of ice on instrumentation, and determination of water surface speeds from tracking river ice floes can contribute to estimate discharge (Beltaos and Kääb, 2014). This possibility is of particular importance for the major Arctic rivers of North America and Siberia, which transport large amounts of freshwater into the Arctic ocean, but the discharge of which is least known for the time of ice break-up – notably the time where annual discharge peaks (Zakharova et al., 2019).

In addition to in-situ measurements and ground-based remote sensing (e.g., Lin et al., 2019), water surface velocities can be mainly retrieved using air- or spaceborne radar interferometry (Romeiser et al., 2007). During periods where rivers carry ice floes, or other visible surface objects, water velocities can be measured using near-simultaneous satellite (or airborne) images, optimally with time separations on the order of minutes (Kääb and Leprince, 2014). Such near-simultaneous imaging of the Earth surface is provided by satellite stereo sensors, where the two or more stereo image partners are by necessity temporally separated by ~1-2 minutes (Kääb and Leprince, 2014). Ice floes (or other floating objects) are then tracked over this time lapse to estimate water surface velocities during the time of image acquisition. Satellite stereo imaging that is useful for this purpose stems either from fixed stereo or agile stereo. (In principle, also satellite video could be used to

track ice floes but has to our best knowledge not been demonstrated yet for this purpose; (d'Angelo et al., 2014; d'Angelo et al., 2016)). Fixed stereo is provided by two or more fixed cameras with different along-track viewing angles; e.g., the ASTER or ALOS PRISM sensors. Agile stereo is provided by one single camera that is rotated during overflight to point repeatedly to the same ground target; e.g., the WorldView or Pleiades satellites. Kääb and Prowse (2011) demonstrated the method deriving river ice and water velocities over reaches of a few tens of kilometres of the Mackenzie and St. Lawrence Rivers, Canada, using both types of satellite stereo images. Kääb et al. (2013) used ASTER fixed satellite stereo to measure and analyse river ice flux and water velocities over a 600 km long reach of Lena River, Siberia. Finally, Beltaos and Kääb (2014) demonstrated how such-derived water surface velocity fields can be used to estimate river discharge. Even if Kääb and Leprince (2014) indicate other seasons and satellite constellations to track river ice floes over short time spans, all the above studies have in common that they (i) use for the most part images during ice break-up, (ii) use dedicated stereo systems, and (iii) use mostly rare and opportunistic acquisitions. Point (i) limits application of the method to one short time period of the year, and (ii) and in particular (iii) prevent the method to be applied operationally and systematically over large reaches of many rivers. The PlanetScope cubesat constellation offers a new, so far not explored possibility to perform systematic worldwide observations of river ice velocities and water velocities indicated by them. The primary aim of the present study is to demonstrate and explore these possibilities, and a secondary aim is to evaluate estimation of water velocities during river freeze-up, instead of during break-up. As the main focus of this study is a methodological one, we do not study in detail selected hydrological, hydraulic, or geomorphological applications that seem possible.

The PlanetScope optical cubesat constellation scans the Earth surface systematically and daily (Figs. 2 and 3) involving overlap of consecutive acquisitions with a time lag of around 1.5 minutes. Such order of time lag is perfectly suited to track floating matter, in particular river-ice floes. PlanetScope thus offers the possibility for systematic daily measurement of water surface velocities, as long as ice floes are present on the water and sky conditions are clear. In this study, we first introduce in more detail the PlanetScope cubesat constellation. After a description of the methods used to track ice floes over minute-scale time-lags, we demonstrate and discuss typical ice-floe conditions suitable for tracking, and derived velocities over a 60 km long reach of Amur River, Siberia, and a 200 km long reach of Yukon River, Alaska. We also discuss the error budget of the measurements in detail. Finally, we draw conclusions on the potential for systematically measuring river ice and water velocities from the PlanetScope constellation and briefly sketch out possible application fields.

## 2 The Planet cubesat constellation

The following descriptions of the PlanetScope constellation and data, and the methods used, are an update and specification of the descriptions given by Kääb et al. (2017). The Planet cubesat constellation, called PlanetScope, consists of small satellites that have a size of 10 cm × 10 cm × 30 cm. Their main components are a telescope and CCD area array sensor. One half of the 6600 × 4400 pixel CCD array acquires red-green-blue (RGB) data and the other half near-infrared (NIR), both in 12 bit radiometric resolution. At the time of writing, the majority of the PlanetScope satellites provides images of about 3.7 m spatial resolution at an altitude of 475 km (delivered as resampled to 3 m; Fig. 1), and a size of individual scenes of roughly 25-30 km × 8-10 km (Planet Team, 2019). While most other optical Earth observation instruments in space acquire in pushbroom geometry (i.e. one-dimensional sensor arrays scanning in orbit direction), the data from the Planet satellites are two-dimensional frame images. Each complete image is taken at one single point in time, has one single acquisition position and one single bundle of projection rays. For comparison, pushbroom sensors integrate an image over a time interval of a few seconds so that acquisition time, position and attitude angles vary throughout an image (Nuth and Kääb, 2011; Kääb et al., 2013; Girod et al., 2015) .

The Planet cubesat constellation consists at the time of writing of around 150 cubesats following each other in two near-polar orbits of ~8° and ~98° inclination, respectively, and an altitude of ~475 km (Fig. 2), imaging the Earth at local morning from both an ascending and descending orbit. The distance along orbit between the cubesats is constructed in a way so that the longitudinal progression between them over the rotating Earth leads to a void-less scan of the surface. The full constellation thus provides sun-synchronous coverage of the entire Earth (except the polar hole) with daily resolution (Fig. 2)(Foster et al., 2015; Kääb et al., 2017). To guarantee this void-less surface imaging at all latitudes and also when satellite positions and pointing angles are not exactly nominal, the swaths of subsequent cubesats overlap in across-track direction by some kilometres (Figs. 2 and 3). Within these swath-overlaps Earth surface targets are imaged twice (sometimes even more) with a time lag of very roughly 1.5 minutes. This time lag is exploited in the present study and constitutes its core principle. The PlanetScope constellation involves also other time lags that are however not considered here (e.g., < 1s between RGB and NIR acquisitions; or a few hours, depending on latitude, between acquisitions from ascending and descending orbits).

During the PlanetScope constellation's technological demonstration phase the cubesats were mostly launched from the International Space Station into an orbit of 52° inclination and ~375 km height (Fig. 2)(Kääb et al., 2017). Data from these satellites form the majority of Planet's cubesat data archive holding for 2016 and into early 2017, before acquisitions from the near-polar sun-synchronous orbits took over. The built-up of the PlanetScope constellation and frequent replacement of its cubesats enables among others fast technological turnover and improvement of the image sensors. As one result, images from the more recent cubesat generations used in this study have typically better radiometric contrast than images from earlier generations.

## 3 Data and methods

Within the swath overlaps and over the corresponding ~1.5 min time lapse we track river ice floes using standard image matching techniques. For image matching purposes, the geometric characteristics of repeat imagery are of particular interest. PlanetScope images are available in different processing levels, and here we use 'analytic' data. 'Analytic' data are radiometrically processed and orthorectified. In this study we do not apply 'unrectified' data, another processing level available, which comes with minimal radiometric processing and in the original central projection. The image orientation from on-board measurements is refined by Planet by matching the scenes onto a global reference mosaic (at the time of writing from Landsat, ALOS and Open Street Map layers) and the images are orthoprojected using a DEM. As for all orthoprojected satellite data, vertical errors in the orthorectification DEM cause lateral distortions in the resulting PlanetScope orthoimages. The size of these offsets is proportional to the DEM error and the off-nadir viewing angle (Kääb et al., 2016; Altena and Kääb, 2017; Kääb et al., 2017). For a worst-case scenario for PlanetScope data (Kääb et al., 2017) a DEM error of 10 m results in orthorectification offsets of around 30 cm in the scene centre and 65 cm at the outer scene margin. For repeat river observations the differential effect of these offsets can be reduced by co-registering the near-simultaneous images using stable points along shorelines. Over the limited width of rivers of a few kilometres in maximum, water surface topography is approximately planar. This makes a first-order polynomial co-registration model sufficient to bring repeat 'unrectified' frame images into overlap. This co-registration procedure will also greatly reduce offsets between the orthorectified 'analytic' images used here as the same DEM is used for both near-simultaneous images (Kääb et al., 2017). Errors in the DEMs used for orthorectifying PlanetScope images are a composite of (i) DEM elevation errors with respect to the real topography at the time of DEM acquisition, and of (ii) real-world elevation changes between elevations at DEM acquisition and elevations at satellite image acquisition. Orthorectification DEMs are by necessity outdated (though generally with limited consequences) unless acquired simultaneously with image acquisition. For river surfaces, the latter elevation deviations will primarily stem from water-level variations between DEM acquisition and image acquisition dates. However, the small field of view of PlanetScope cubesats and the resulting small sensitivity to orthorectification DEM

errors, the frame geometry of the PlanetScope cameras, and the accessibility of unrectified images, if needed, all contribute to minimize topographic distortions.

Bright ice floes on a dark water surface constitute features of strong visual contrast and tracking them over short time intervals is a particularly easy task for image matching algorithms. For matching the repeat PlanetScope data we thus use a standard method, normalized cross-correlation (NCC), solving the cross-correlation in the spatial domain and reaching sub-pixel accuracy by interpolation of the image (Kääb and Vollmer, 2000; Debella-Gilo and Kääb, 2011b; Kääb, 2014). NCC solves for translations between corresponding image elements. We apply the software Correlation Image Analysis software (Kääb, 2014), but established scripts or routines for normalized cross-correlation between images exist for many programming languages. As the tracking of ice floes over short time intervals represents little challenge for image matching, we expect that other image matching methods (Heid and Kääb, 2012; Lin et al., 2019) bring no substantial advantage. Over longer time intervals, though, or strong horizontal water turbulences such as backwaters, ice floe rotation over time will get significant so that image matching methods that are able to model feature rotation in addition to translation could be advantageous (Debella-Gilo and Kääb, 2012). The matching window sizes used in this study for the PlanetScope data are 30×30 pixels (90×90 m) as found roughly optimal from a few tests. Tests with different window sizes are, though, not the focus of this study (Debella-Gilo and Kääb, 2011a). Measurements with a correlation coefficient smaller than 0.7 are removed and no other post-processing is applied (Kääb et al., 2017).

For comparing and supplementing our results based on Planet cubesats, we also use data from other satellites. A Landsat 8 scene of 16 September 2013 (i.e. ice-free conditions) is employed to automatically delineate the river water surface over our Yukon River study reach. Indexes used for the purpose of mapping water areas from multispectral satellite data are typically based on the reflectance contrast of water between blue (high reflectance) and near-infrared wavelengths (low reflectance) (McFeeters, 1996; Pekel et al., 2016). For our study site and conditions, we find however that the contrast between the blue and thermal infrared Landsat bands is larger than the blue vs. infrared contrast because of high suspended sediment concentration that increases the near-infrared reflectance and thus reduces the contrast to reflectance at blue wavelengths. To increase index sensitivity compared to the often used normalized difference indexes (McFeeters, 1996), we apply a band ratio. River outlines were thus obtained from a raster-to-vector conversion of a noise-filtered (3×3 median filter) and thresholded band ratio image (Paul et al., 2002) between the blue and thermal infrared bands of Landsat 8. The Landsat 8 blue band has 30 m spatial resolution, and the thermal infrared bands are also provided at 30 m resolution, though originally taken at 60 m resolution.

For one of our Planet cubesat acquisition pairs over Yukon River, a Sentinel-2 scene exists taken with 1 h time difference. Sentinel-2 multispectral data have a spatial resolution of up to 10 m (Drusch et al., 2012; Kääb et al., 2016). We visually identified the position of a number of large ice floes corresponding between the Planet and Sentinel-2 images and measured the associated displacement along the river to estimate average velocities over the 1 h time period.

In order to compare the velocity retrieval from Planet cubesat data to a method used earlier for the same purpose we measure short-term ice floe displacements over the Yukon River reach also from an Advanced Spaceborne Thermal Emission and Reflection Radiometer (ASTER) stereo strip. The ASTER fixed satellite stereo, taken at 15 m spatial resolution and in near-infrared, implies a time lapse of around 55 s between the two images of a stereo pair that can be exploited to track ice floes in a way very similar to the Planet cubesat images. The exact procedures, performance, and accuracies are presented in Kääb et al. (2013). As a speciality, satellite vibrations (so-called jitter) were modelled and corrected for when using the ASTER data. The results presented here for the Yukon River are based on an especially tasked ASTER acquisition, and have not been published before.

The river flow results of this study are presented as simple maps of measured velocity vectors or magnitudes, or as longitudinal profiles of water flow speed and derived parameters. For the latter we have to average the velocity vectors along the river reach. For that purpose we move a running window, which is 4 km long in reach direction and has infinite width, in 100 m steps along the mean direction of a study reach. The window length of 4 km and step size of 100 m are experimentally chosen for our study sites to smooth the measurements substantially but at the same time leave enough details.

At each window position the number of measurement grid points within the river mask from Landsat 8 data, and average (or median) speeds and directions for the velocity measurements within the correlation threshold are computed. Dividing the number of grid elements within the river mask by the window length gives then an approximate river width for each step. This river width is then corrected for the deviation between mean flow direction per window step against the overall mean direction of the river reach studied, essentially rotating the window at each step to align with the actual flow direction. (Note that other procedures exist that are more specialised for estimating river width without flow vectors available (Allen and Pavelsky, 2018)). The surface area flux is then the multiplication of average river speed and (corrected) width for each window step.

## 4 Results

### 4.1 River ice conditions

Figure 4 illustrates a small subset of typical river ice conditions in Planet images that are suitable for tracking ice floes or ice features, and estimating water velocities. During break-up we find predominantly smaller ice floes with very variable densities of ice-floe cover (Fig. 4, right column). During freeze-up we find typically bigger ice floes and a more equal distribution of ice-floe cover density over the river surface. This simple description of differences between freeze-up and break-up ice conditions is an overall and qualitative one based on a substantial, though, visual exploration of Planet archive holdings, but a range of exceptions and natural variations certainly exist. Our extensive searches in the Planet image archive suggest clearly that during river freeze-up the ice conditions that are suitable for tracking are more constant over time and they stretch over longer time periods (up to several days or even a week, roughly) compared to break-up conditions. Break-up ice conditions that are suitable for tracking last typically only one or several ice pulses of a few days in maximum, often just a day or two. This makes it more probable to acquire/find images of suitable ice conditions during freeze-up than during break-up. On the other hand, though, for the northernmost latitudes the freeze-up period reaches into the season of low sun angle, where Planet cubesats (and other optical satellite instruments) do not acquire data anymore due to too little solar radiation reaching and reflecting at the Earth surface. Still, our clear overall impression is that it is typically easier to find Planet images that are suitable for tracking ice floes over freeze-up than over break-up periods.

### 4.2 Amur River, Siberia

For a first example of river ice velocities retrieved from near-simultaneous Planet cubesat images we mosaic two overlapping sets of 12 scenes each into two image strips covering a ~60 km long reach of the Amur River near the city of Komsomolsk-on-Amur, eastern Siberia. The image-strip pair was acquired on 1 November 2016 (~22:46 UTC) from an International Space Station (ISS) orbit, with a 73 s time lapse. Figure 5a shows one of the two image strips as an infrared false colour composite. The freeze-up river ice conditions during the acquisition were close to perfect for matching velocities. Ice floes densely covered most of the water surface, but were at the same time for most areas not colliding with each other so that ice floe velocities should to a large extent indicate water velocities at their locations. Ice floe collisions would transfer additional lateral forces that overly the downstream drag by the river flow. Ice conditions on 1 November 2016 are shown in Fig. 6a. The diameter of visible ice floes ranges for the most part from around one pixel (3 m) to roughly

100 m, with a number of individual floes reaching up to 200-300 m. Figure 5b shows the magnitudes of the velocities derived, with maximum speeds of 1.7 m s$^{-1}$ close to the lowest elevation of the river reach investigated (right margin of Fig. 5b). The displacement measurements were done within a manually digitized polygon roughly delineating the river floodplain around the river. The same correlation threshold of 0.7 was applied to all measurements, both on the river and outside. Successful displacements (i.e. measurements that passed the correlation threshold) are dense on the river but sparse on the floodplain surrounding the river as the surface there seems to consist mostly of homogenous shrubs that offer little visual contrast to match at the image resolution of 3 m. The images used in this example over Amur River stem from an early generation of Planet cubesats providing images with less good radiometric contrast compared to images from current Planet cubesats (see section 2). In addition, contrast is reduced by the low sun angle during the acquisition. Despite these two complications, matches of ice floes seem robust with accuracy and reliability little affected because the bright floes offer particularly strong visual contrast against the surrounding dark water surface. In summary, the sparse displacements surrounding the river (scattered blue results in Fig. 5) reflect on the one hand the lack of good visual contrast to match between the two images on the floodplain. On the other hand, the small magnitude of these spares displacements confirms that the two images co-register well. Figure 6 shows a detail (rectangle in Fig. 5) of the original velocity vectors measured. Grid spacing of the vectors is 75 m.

Figure 7 presents the longitudinal profile of speeds for the 1 November 2016 data set, together with the river width automatically derived from the velocity vectors. Further we also compute the 2-dimensional (2D) surface area flux as a function of transverse velocity profiles. As an example for interpretation of the longitudinal profile, at ~25 km 2D surface area flux is relatively low, suggesting under mass conservation that the Amur River should be relatively deep at this part of the reach. In contrast, the river should be on average relatively shallow at, for instance, ~55 km. Interpretation of the longitudinal profile is influenced by the multi-branch geomorphology of the Amur River reach studied. In the individual speed measurements (grey dots in Fig. 7) branches become expressed by clusters of different speeds at the same reach section. For instance, at ~15-20 km speeds on one branch are around 0.7 m s$^{-1}$, on the other branch up to 1.2 m s$^{-1}$. Two clusters with different mean speeds on different branches are also well visible at around 30 km.

## 4.3 Yukon River, Alaska

For a second case study we chose a ~200 km long reach of the Yukon River, Alaska (Fig. 8). Over this reach, the overall river azimuth coincides with the azimuth of the near-polar descending orbit of the Planet cubesats. We mosaic sequences of around 25 scenes each to obtain two image strips for 16 May 2017 (~21:12 UTC) with 15 s time lapse, and two images strips for 4 November 2018 (~21:30 UTC) with 171 s time lapse. Typical ice conditions for these acquisitions are demonstrated in Figs. 4h and g, respectively. The diameter of visible ice floes on the 4 November 2018 images ranges between around one pixel (3 m) and 100 m and more. There are many large ice floes of up to around 100 m in diameter. Larger ice floes of up to around 200 m can be found but are less frequent than on the Amur River images (section 4.2). For 16 May 2017, the ice floe diameters are significantly smaller and typically not exceeding a few pixels. The velocity magnitudes derived are shown in Figs. 8c and e, speed differences between them in Fig. 8d, and a detail of these three items in Fig. 9. For comparison to a method that was used earlier, we add river ice speeds derived for 13 May 2014 from a strip of ASTER stereo pairs (i.e. 55 s time lapse) following the method by Kääb et al. (2013) and compute differences to the 16 May 2017 Planet data set (Figs 8a and b). On 13 May 2014, river ice cover was comparably sparse and subsequently also the density of successful velocity matches. The freeze-up conditions of 4 November 2018 clearly offered the most complete cover by river ice floes and thus the most complete velocity field. The river outlines used in Figs. 8 and 9 were obtained from a Landsat scene of 16 September 2013 as described in section 3. Visually, these outlines represent the actual outlines of May 2014 and May 2017 very well, without significant changes over time. At shallow river parts, outlines of November 2018 (i.e. low water

conditions) were of course more narrow than for September 2013. The outlines produced here are however only used for visualisation, and initial result segmentation into classes "river" and "outside river" for accuracy assessment on stable ground.

The closest river discharge measurements to our Yukon River reach are done at Pilot Station (no. 15565447), some 300 km downstream from the lower end of the reach studied. For 13 May 2014, 16 May 2017, and 4 November 2018 discharge estimates at Pilot Station are 11,383 $m^3$ $s^{-1}$ , 8,410 $m^3$ $s^{-1}$ , and 5,437 $m^3$ $s^{-1}$ respectively. Taking into account the distance between the reach investigated and Pilot Station, we also give the discharges 3 days later: 13,450 $m^3$ $s^{-1}$ , 11,213 $m^3$ $s^{-1}$ , 4,927 $m^3$ $s^{-1}$ for 16 May 2014, 19 May 2017, and 7 November 2018, respectively. Similar to the discharges also the surface velocities measured for 13 May 2014 are higher than for 16 May 2017, and the latter ones are higher than for 4 November 2018, as can be seen from panels Fig 8b and d. Mean speed of 4 November 2018 is 0.80 $m^3$ $s^{-1}$ , and 1.35 $m^3$ $s^{-1}$ for 16 May 2017. Due to the sparse coverage by successful measurements in the ASTER data (Fig. 8a), only few differences can be computed to the Planet data (Fig. 8b). The differences between the two Planet data sets (Fig. 8d) are much denser, demonstrating the advantage of the high-resolution Planet cubesat data in combination with the denser coverage by ice floes during freeze up. Speeds between 19 May 2017 and 7 November 2018 vary both on longitudinal average (Fig. 10b) and across the river (Figs. 8d and 9). In future applications, these measured spatio-temporal variations of surface water speed could be analysed in combination with known bathymetry and/or hydraulic formula.

Figure 10a shows the longitudinal profile of speeds for the 4 November 2018 data set, together with the river width automatically derived from the velocity vectors. Further, we also compute the 2-dimensional (2D) surface flux as a function of transverse velocity profiles. As an example for interpretation of the longitudinal profile, at ~80 km 2D surface flux is relatively low, suggesting under mass conservation that the Yukon River should be relatively deep at this part of the reach. In contrast, the river should be on average relatively shallow at, for instance, ~120 km.

From a similar profile of river surface speed along 400 km of the Lena River during 27 May 2011, Kääb et al. (2013) found a striking peak in the power spectrum of river surface speed at 20.8 km. For the Yukon River profile Fig. 10 we find a somewhat less prominent but still significant peak in the power spectrum of speeds at 20.5 km. The similar number for both river reaches might point to similar processes and parameters for the development of the respective river morphologies (Lanzoni, 2000a, b). Kääb et al. (2013) provide some more discussion on the 20.8 km wavelength speed-variations including comparison to a topographic profile.

Profile Fig. 10b compares river surface speeds and river widths of 16 May 2017 and 4 November 2018. The four data sets are consistent in the sense of mass conservation; higher discharges in May 2017 compared to November 2018 (see above discharges for Pilot Station) correspond to a combination of larger widths and higher surface speeds. For instance, at sections where river width is significantly larger in May 2017 than November 2018, speed differences between May 2017 and November 2018 are smaller (e.g. at ~60, 110 or 170 km). Conversely, at sections with relatively small changes in river width, surface speeds change more (e.g. at ~30, 90, or 130 km).

### 4.4 Error budget

The error budget for individual river surface velocity measurements consists of three main components: (i) offsets in absolute georeference of a set of repeat images, (ii) relative distortions and offsets between repeat images, and (iii) errors from matching of features between the repeat images. The first category, the uncertainty of the absolute georeference, stems mostly from matching the Planet images onto a reference image. This step is part of the Planet in-house processing and is to our experience typically accurate in the order of one pixel or less, but can be larger for partially cloud-covered or snow-covered scenes. Failure or gross uncertainties of this georeference refinement step and subsequent gross georeference errors

are flagged by Planet in the image meta-data. To our best knowledge, an absolute georeference accuracy of a few meters or pixels for the locations of derived velocities should not be a problem for most applications, in particular when considering that the derived velocities anyway represent a window of several tens of metres (here 90 m × 90 m). The second category of uncertainty, (ii), distortions and offsets between the images matched, can be minimized by co-registration, which is typically

possible with sub-pixel accuracy. This uncertainty source is not necessarily eliminated for small-scale higher-order distortions (see section 3) that differ between the stable ground used for co-registration (river shore, flood plain, etc.) and the actual river surface. The parts of this second error component that are not eliminated by image co-registration mix with the third error category, that is the actual matching accuracy for the stable ground or river ice features, respectively (iii). This relative matching accuracy between already co-registered images defines the uncertainty of the actual displacements or

velocities derived, and we thus consider it here as the error component of largest interest (Kääb et al., 2013) and focus on it in more detail in the following.

Uncertainties of individual velocity measurements or outliers (our above error component iii) stem from uncertainties in definition of river-ice features over time, i.e. how sharp can features be matched that change over time and how (precisely) is a displacement between slightly modified features defined. This error component includes the representativeness of

displacements matched using a 90 m × 90 m window for actual point-wise velocities, and the degradation of the matching accuracy by rotation or deformation of river-ice features over the minute-scale time lapse exploited here (Kääb et al., 2013). We estimate the accuracy of our river ice velocity measurements in three ways: (1) inferring from previous studies, (2) stable ground matches, and (3) variance of velocities within homogenous parts of the derived flow field.

(1) Based on ASTER data over the Lena river, Kääb et al. (2013) suggest for most optimal imaging and ice conditions a

displacement accuracy of up to 1/8 of a pixel, which would in our case translate to about ± 0.4 – 0.5 m (or 0.005 m s$^{-1}$ for a 90 s time lapse).

(2) Based on about 27'000 matches on the floodplains around the rivers investigated in this study we obtain a mean displacement dx of -0.1 ± 0.5 m, dy of -0.2 ± 0.6 m, and mean displacement length (Pythagoras of individual dx and dy) of 0.4 ± 0.6 m. Besides a good co-registration accuracy of around 0.2 m (i.e. about 1/15 of a pixel), our stable ground tests

suggest thus an accuracy of individual velocity measurements of ±0.6 m (1/5 of a pixel; 0.007 m s$^{-1}$ for 90 s). This latter number agrees well with the accuracy estimates for co-seismic displacement measurements from repeat Planet data of 1/4 of a pixel (Kääb et al., 2017). Figure 10c shows a longitudinal profile of stable ground matches (in m s$^{-1}$; black dots) for the 4 November 2018 data. The stable ground median speed is 0.02 m s$^{-1}$, and the mean 0.03 m s$^{-1}$. Similar results are found for 16 May 2017. These values can be considered an upper limit for the accuracy of ice floe measurements as the river ice floes

offer better visual contrast for the matching than the areas surrounding the river (see section 4.2), and the image areas outside of the river are likely subject to larger topographic distortions than the river surface (see section 3).

Finally, (3), variations of velocities within homogenous parts of the derived flow fields, i.e. the standard deviation of means over such parts of the flow fields, range in our tests between ±0.3 m for the shortest time lapse in our study (15 s; translating to 0.02 m s$^{-1}$) and ±3 m for our longest time lapse (171 s; 0.02 m s$^{-1}$). Especially for the longer time lapses, deformations of

the rive ice features matched and rotations of individual ice floes certainly degrade the actual matching accuracy. From all our above three approaches we suggest thus as a rule of thumb an accuracy in the order of ±0.01 m s$^{-1}$ for individual river ice velocities derived from near-simultaneous PlanetScope data. Note that this accuracy improves following standard error propagation rules, once individual velocities are averaged, for instance for cross-sectional or longitudinal means.

As another possible indicator of measurement quality, Fig. 10c shows the percentage of successful matches on the river.

Clearly, this percentage is much higher for the 4 November 2018 freeze-up conditions than for the 16 May 2017 break-up conditions on Yukon River. This indicator can be used in several ways, for instance for masking out the results for reach

sections with low values, for developing weighting of the above nominal accuracy, or for analysing unwanted dependencies between results and measurement density. The stable ground matches (dots in Fig 10c) also exhibit errors in co-registration. For the 4 November 2018 data, a small co-registration problem can be seen at ~140-160 km with elevated speeds. Kääb et al. (2013) demonstrate a procedure to correct such offsets.

## 5 Discussion, conclusions, and outlook

In this study we exploit the fact that the cross-track overlaps of the swaths of subsequent PlanetScope cubesats (Figs. 2 and 3) produce near-simultaneous optical acquisitions, separated by ~90 s. Over this time lapse we track river ice floes and use them as indicators for water surface velocities. Planet cubesats scan the entire land surface of the Earth at daily repeat and with ~3 m spatial image resolution. Our study shows that these data substantially extend the possibilities to measure river ice and water surface flow from near-simultaneous optical satellite data. Over many rivers that carry river ice, ice floes can be tracked during freeze-up and/or break-up with accuracies in the order of $\pm 0.01$ m s$^{-1}$. Freeze-up conditions appear to be particularly well suited for this work due to the longer time periods where ice floes are present, and more favourable types and densities of ice floes.

We find three main obstacles when applying the method. By constellation design, the PlanetScope cross-track overlaps (never intended for measuring minute-scale changes and motions!) cover not the entire Earth surface but only parts of it, depending on latitude, for instance 2/3 of a cubesat swath at 65° North (Fig. 3). Second, cloud cover seems rather typical for the river freeze-up and break-up seasons, and considerably complicates the acquisition of suitable Planet cubesat data – as for any optical satellite instrument. Third, freeze-up for some northern-most rivers or river reaches seems to happen during sun angles that are too low to acquire suitable images. As a very rough guess from our Planet cubesat archive searches, we estimate that there is for a given river location a 50% chance to get a least one cubesat image per year with drifting ice visible. The chances that the river location is then included in the swath overlap from a subsequent cubesat is lower, and considerably lower for the river being covered several times (per year or in total) by overlaps that enable tracking. Despite these limitations, though, the tracking of river ice in near-simultaneous Planet cubesat data substantially increases the possibilities for deriving surface velocities on cold-region rivers compared to the very few occasional optical stereo acquisitions suitable for the same purpose.

The strong visual contrast provided by the bright ice floes on the dark water surface, together with the short time lapse of around one minute exploited here, lead to little other motion components than translation and represent quite optimal conditions for image matching. Therefore, and because the focus of our study lies on evaluating the potential of the Planet cubesat constellation rather than on the image matching algorithm, we used standard normalized cross-correlation (NCC) as tracking method. Future work could test if other tracking methods have advantages against NCC for tracking ice floes in near simultaneous satellite images. In particular for sea-ice tracking, other matching procedures are used that are optimized to work on sequences of low-resolution satellite data with time lags of hours to days (e.g., Lavergne et al., 2010; Petrou and Tian, 2017). An overview and assessment of state-of-the-art image-based tracking approaches for water flow measurements, where some are certainly relevant for near-simultaneous Planet cubesat data, is given in Lin et al. (2019).

The parameters provisionally chosen for the moving windows to compute longitudinal flow averages (4 km length, 100 m step width) could easily be adjusted. Our visualisations turn out to be little sensitive to the exact choice of window parameter values. For the long river reaches studied here, the mean river direction defining the initial window orientations is almost identical with the orbit azimuth. The image matches on the floodplain outside the rivers can thus easily be transformed into their satellite along-track and cross-track components, which is a preferred coordinate system to analyse the geometric performance and errors in satellite data (Kääb et al., 2013). In the present study we do not find geometric errors of concern,

such as for instance satellite jitter, which Kääb et al. (2013) find and correct for a similar study based on another satellite data type.

As we use our longitudinal averaging procedure only for visualisation purposes, it is not optimised for specific applications such as estimating river width, discharge, or parameters of river morphology and flow. In particular, larger voids in the measured velocity field, due to low correlation coefficients, will bias the flow averages per window step. This effect seems strongly reduced for freeze-up conditions as the coverage by ice floes during these conditions appears to be typically much more complete compared to break-up conditions (see section 4.1). River areas without ice floes lead only to voids in the measurements if they are larger than the matching window size (here 30×30 pixels; 90×90 m) in at least one dimension as the matching algorithm used (NCC) is not sensitive to where the matched features are located in the window. A first measure to indicate problems from voids in the velocity field in the profiles is to plot the percentage of void pixels per window position (Fig. 10c). Smaller voids could be filled, whereby the measured velocities enable application of a directional interpolator. Or, the matching window sizes can be automatically adapted to the distribution of ice floes – small windows for dense ice floes and larger ones for sparse ice floes (Debella-Gilo and Kääb, 2011a). The effect of voids on derived parameters can be tested by simulating voids for a rather complete data set (e.g., Yukon River 4 November 2018) from actual voids in another data set (e.g., Yukon River 16 May 2017) (McNabb et al., 2019).

Another effect to be taken into account is the influence of river branching on the averages. Again, also treatment of this effect depends much on application and parameters of interest. For instance, the mean flow speed and surface area flux that we compute are not affected, while making connections between our surface flow measurements and river discharge would require taking branching into account. An initial simple procedure for that purpose would be to intersect the moving window at each step with the river outlines and compute the flow averages for each intersection area separately.

Although not exploited closer in this study, we would like to note the existence of a Sentinel-2 scene of 4 November 2018, taken about one hour after the Planet scenes over Yukon River. Due to this large time lapse between the Planet and Sentinel-2 scenes and the related large displacements and deformations/rotations of river ice features, traditional image matching methods that solve only for translations are complicated, but manual tracking of distinct floes is still clearly possible. Tests show good agreement between the speeds derived over 1 h and those over 171 s. The fact that most Planet cubesats, Sentinel-2A and 2B, and Landsat7 and 8 are on similar orbits can thus create additional opportunities for tracking river ice movement, for investigating short-term changes in river ice cover and speed, and for additional or combined multispectral mapping and analysis together with the Planet cubesats.

As demonstrated here for a 200 km long reach of the Yukon River, remotely sensed water velocities over long reaches might offer improved insights in river morphology. For instance, we find a variation of water speeds of ~20-21 km wavelength for the Yukon River (and the Lena River; Kääb et al. (2013)) that could be compared to according wavelengths found from laboratory experiments and models on bar formation (Lanzoni, 2000a, b).

A major purpose of satellite-based river observations is to estimate discharge in order to spatially or temporally complement the sparse in-situ measurements available from gauging stations (Beltaos and Kääb, 2014; Bjerklie et al., 2018; Zakharova et al., 2019; and many others, see references in the cited ones). River velocities from the approach demonstrated here can offer an additional type of input measurement (or a possibility for independent comparison) when linking satellite-based measurements of river height and slope from altimetry data, and measurement of river surface from optical (Allen and Pavelsky, 2018) or radar images, to standard discharge equations (Bjerklie et al., 2018; Zakharova et al., 2019). Such satellite data are available over large regions (Allen and Pavelsky, 2018) and fit thus well to the river velocities as derived by our approach. Satellite-altimetric river heights will even improve in the (near) future through the new Sentinel-3 and high-

resolution ICESat-2 missions (Brown, 2019), and the upcoming SWOT mission (Durand et al., 2010). And Landsat 8 and Sentinel-2 together offer sub-weekly repeat to measure parameters such as river width.

As further outlook, the water mapping opportunities from the daily repeat Planet data (Cooley et al., 2017) together with opportunities to measure ice velocities from them, as demonstrated here, could aid detecting ice jams and related flooding (Fig. 1), as well as provide better understanding of the mechanisms involved in ice jam formation. The damages from ice jam floods cause annual economic costs on the order of several hundred millions EUR per year in North America and Siberia (Prowse et al., 2007; Rokaya et al., 2018b, a). Finally, while substantially fewer in number, we speculate that near-simultaneous overpasses in tropical and temperate rivers could similarly be exploited, tracking sediment or floating matter in place of ice (Kääb and Leprince, 2014).

**Code availability**

The image matching code used for this study (Correlation Image AnalysiS, CIAS) is available from http://www.mn.uio.no/icemass.

**Data availability**

Sentinel-2 data are freely available from the ESA/EC Copernicus Sentinels Scientific Data Hub at https://scihub.copernicus.eu/, Landsat 8 data from USGS at http://earthexplorer.usgs.gov/, ASTER data from https://earthdata.nasa.gov/ , Yukon River discharge data from https://waterdata.usgs.gov. Planet data are not openly available as Planet is a commercial company. However, scientific access schemes to these data exist (https://www.planet.com/markets/education-and-research/).

**Author contribution**

A.K. developed the study, did most of the analyses and wrote the paper. B.A. supported the analyses and edited the paper. J.M. helped with data acquisition, technical details to the Planet constellation and data, and edited the paper.

**Competing interests**

A.K. and B.A. declare that they have no competing interests. J.M. was program manager for impact initiatives at Planet. He influenced in no manner the results or conclusions of the study.

**Acknowledgements**

Special thanks are due to two anonymous referees for their detailed and constructive comments, and to the editor of this paper. We are grateful to the providers of data for this study; Planet for their cubesat data via Planet's Ambassadors Program, ESA/Copernicus for Sentinel-2 data, USGS for Landsat 8 and river discharge data, and NASA and the ASTER science team at JPL for ASTER data. Special thanks are due to Michael Abrams for tasking the May 2014 ASTER acquisitions over the Yukon River. The work was funded by the European Research Council under the European Union's Seventh Framework Programme (FP/2007-2013) / ERC grant agreement no. 320816, the ESA project Glaciers_cci (4000109873/14/I-NB), the ESA Living Planet Fellowship to B.A. (4000125560/18/I-NS), and the ESA EarthExplorer10 Mission Advisory Group.

**Figures**

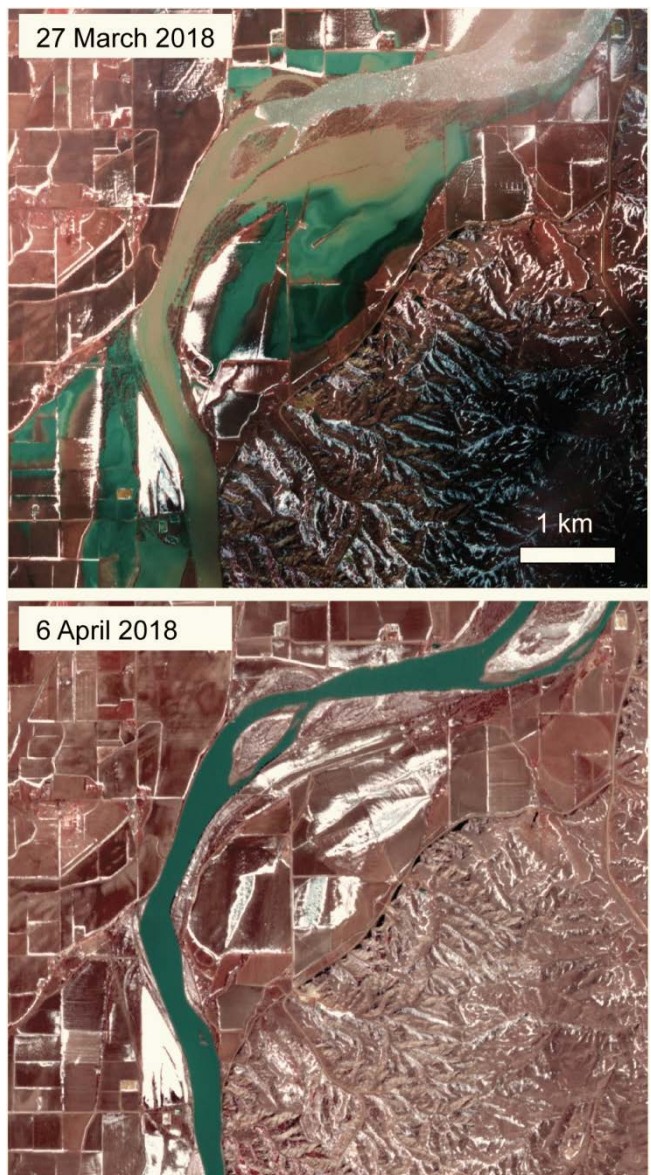

[1 column wide]

5  **Figure 1: Planet images over an ice jam on Yellowstone River at Sidney, NE, USA (47.75° N, 104.09° W). The river flows from bottom to top (North). Top: ice jam (top) and associated flooding. Bottom: after break of the ice jam.**

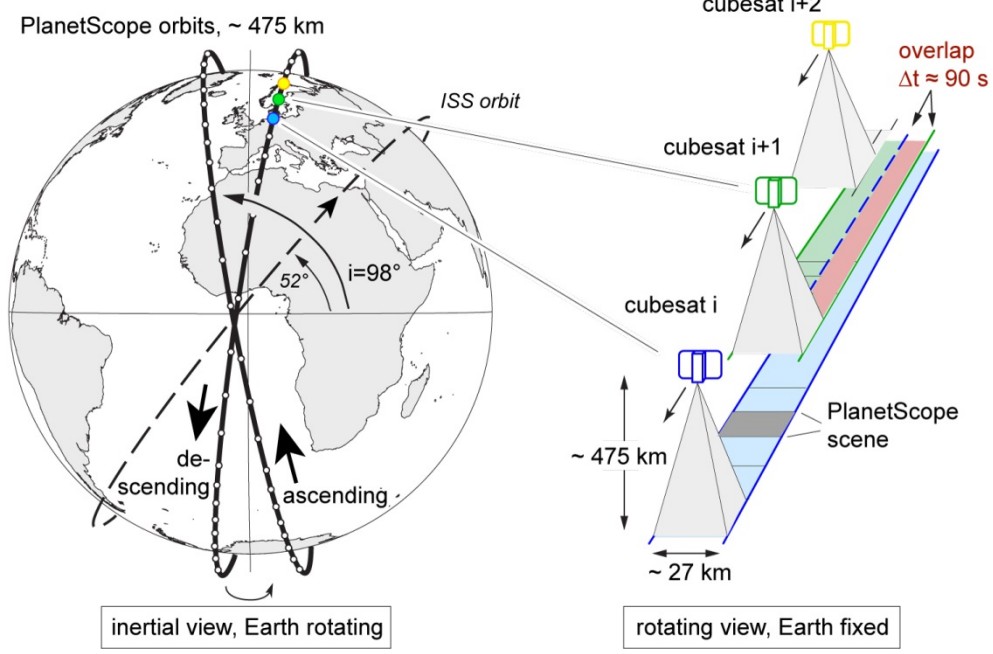

[1.5 columns wide]

**Figure 2: Planet orbits. Left, inertial view: final PlanetScope descending and ascending orbits (bold) and ISS test-bed orbit (dashed). Cubesat positions (white dots on the orbits) are only schematically indicated. Right, rotating view: scheme of complete**
5  **scan of the Earth surface by successive PlanetScope cubesats in the same orbit producing a time lapse of around 90 s over the swath overlaps.**

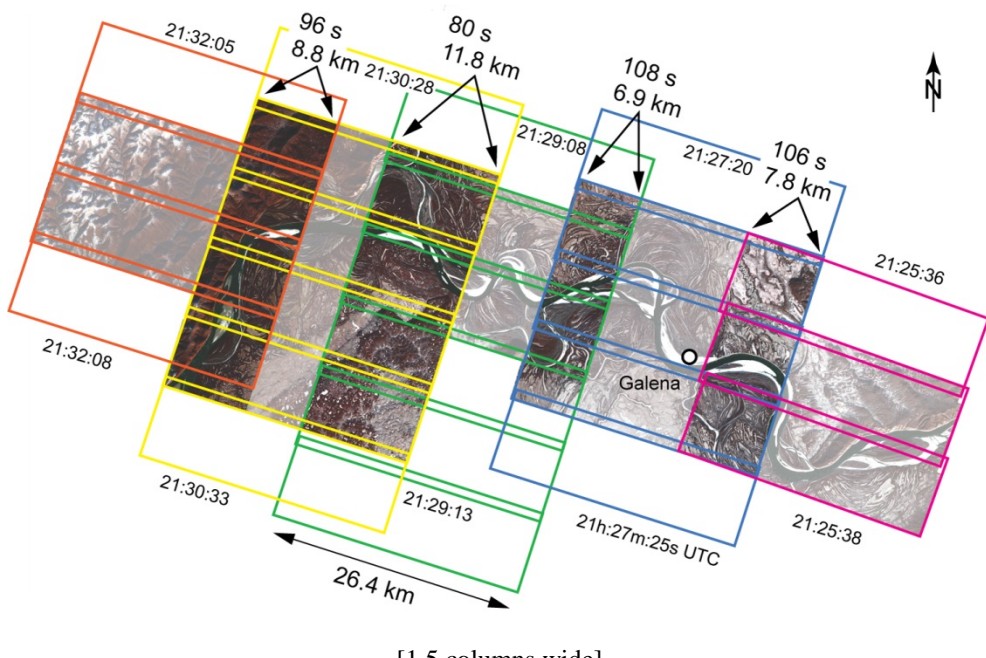

[1.5 columns wide]

**Figure 3: Typical PlanetScope acquisition pattern on a cloud-free day during freeze-up (28 October 2018) over the Yukon River at**
10  **Galena, Alaska (64.75° N, 157° W). Each colour indicates one satellite swath with individual scenes. Non-dimmed image parts indicate scene sections where two images with time lapse between them exist and river ice floes can be tracked. Time lapse and width of the overlaps are given together with UTC time of the acquisitions.**

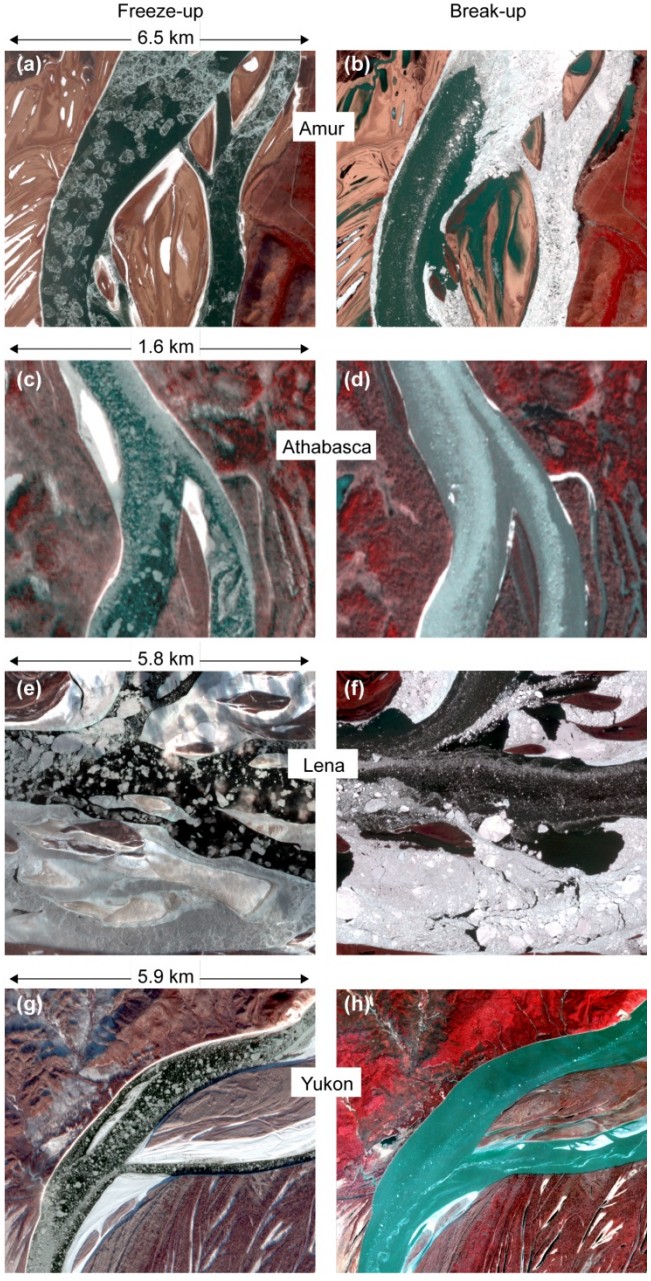

[1 column wide]

**Figure 4: Typical river ice conditions in Planet imagery (shown in infrared false colour) that are suitable for tracking ice floes to estimate water velocities. Left column: during freeze-up; right column: during break-up.**

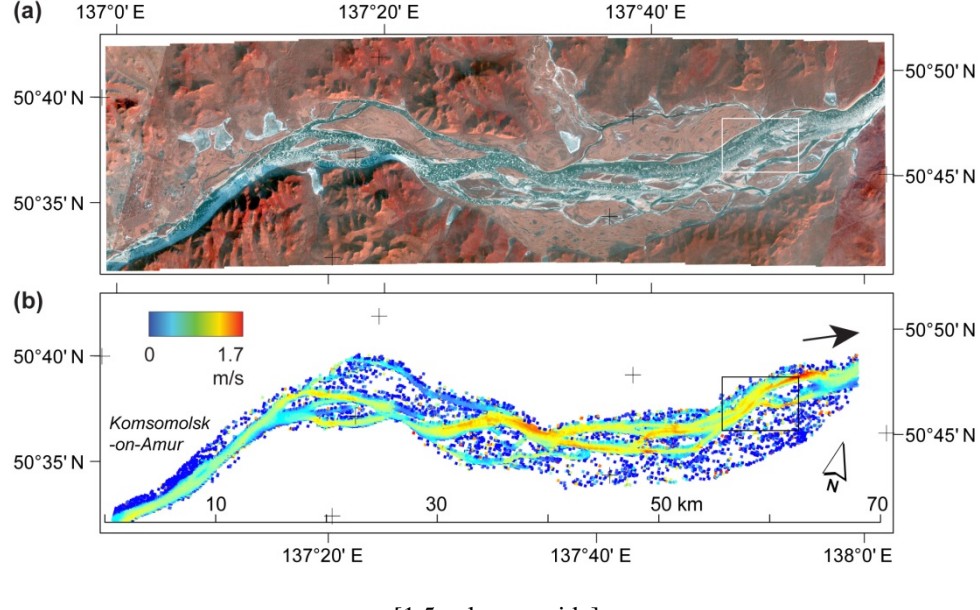

[1.5 columns wide]

**Figure 5: Amur River near the city of Komsomolsk-on-Amur, Siberia (lower left corner). River surface velocities of 1 November 2016 are tracked over a 73 s time lapse between overlapping Planet cubesat images. (a) False colour composite of one of the image strips, (b) derived surface speeds. Overall flow direction is from left to right. The small rectangle marks the location of detail Fig. 6.**

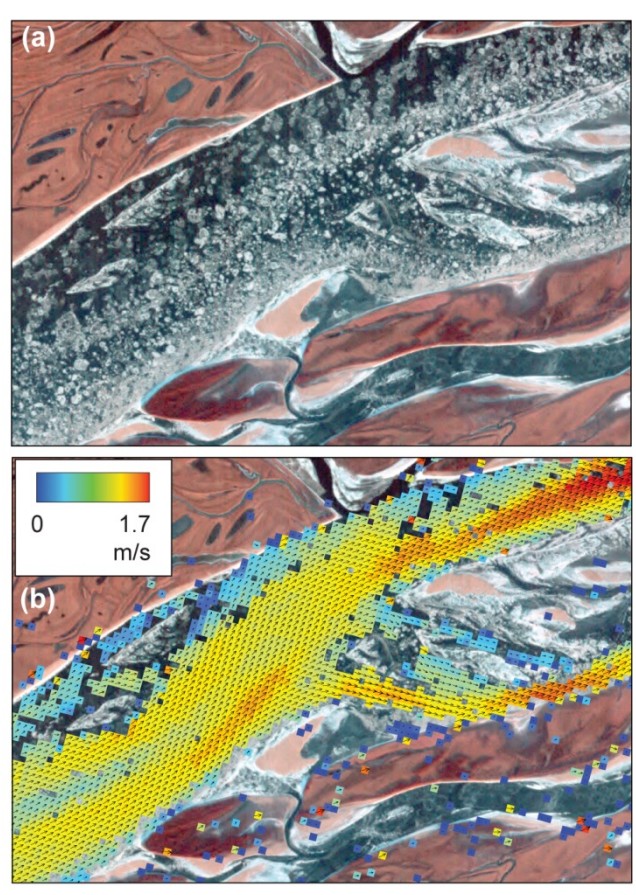

[1 column wide]

**Figure 6: Detail of Fig. 5 (small rectangle in Fig. 5). (a) Planet cubesat image of 1 November 2016, (b) original matched surface velocities after thresholding of the correlation coefficient. Grid spacing of vectors is 75 m. Matching results are given in colour-coded speed and with velocity vectors superimposed. Maximum speed 1.7 m s$^{-1}$.**

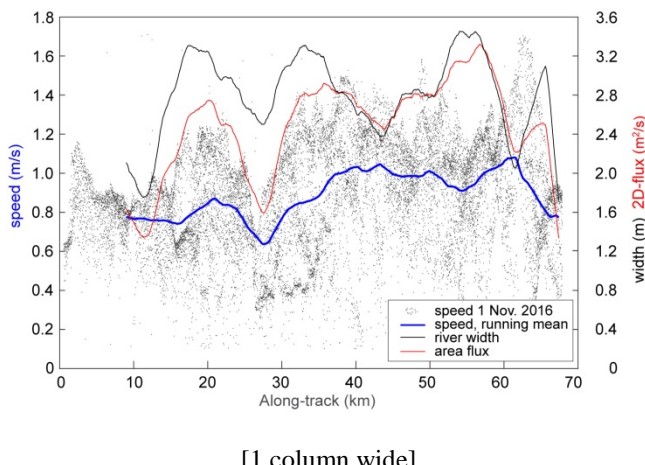

[1 column wide]

**Figure 7: Longitudinal profile of mean speeds and river widths derived from near-simultaneous Planet cubesat images of 1 November 2016 over a reach of Amur River (Fig. 5). Small dots: individual speed measurements; blue line: 4 km running mean of individual measurements; black line: river width from velocities (running mean); red line: surface area flux as product of cross-sectional average speed and river width (running mean).**

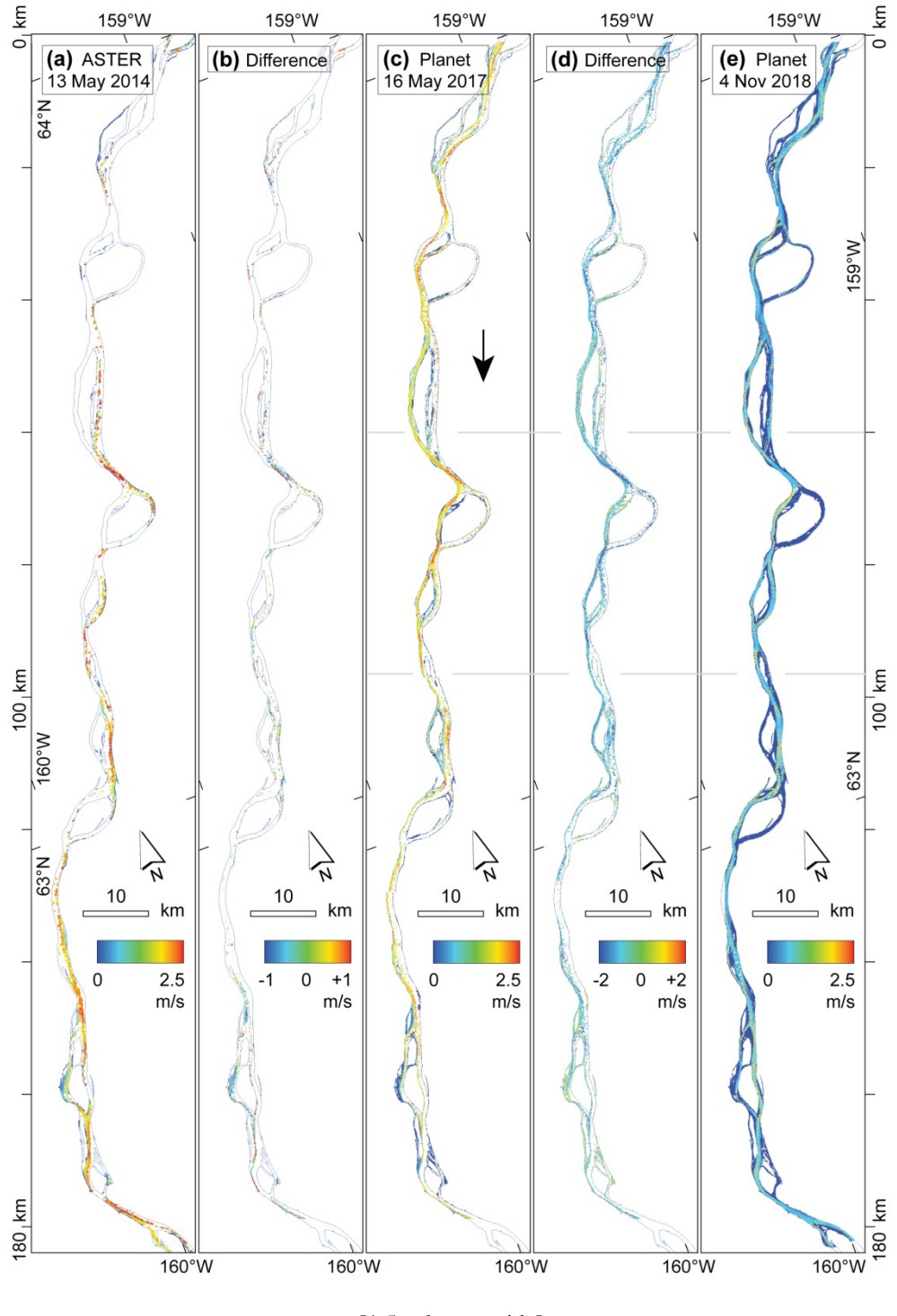

[1.5 columns wide]

**Figure 8: Surface velocities on Yukon River, Alaska, from near-simultaneous satellite images. Flow direction roughly from north to south, top to bottom of figure. Velocities from (a) an ASTER stereo pair of 13 May 2014 (55 s time lapse), (c) two Planet cubesat image strips of 16 May 2017 (15 s), and (e) of two Planet cubesat image strips of 4 November 2018 (171 s). Panels (b) and (d) show the differences (c)-(a) and (e)-(c), respectively. The horizontal grey lines in panels (c) to (e) indicate the detail shown in Fig. 9.**

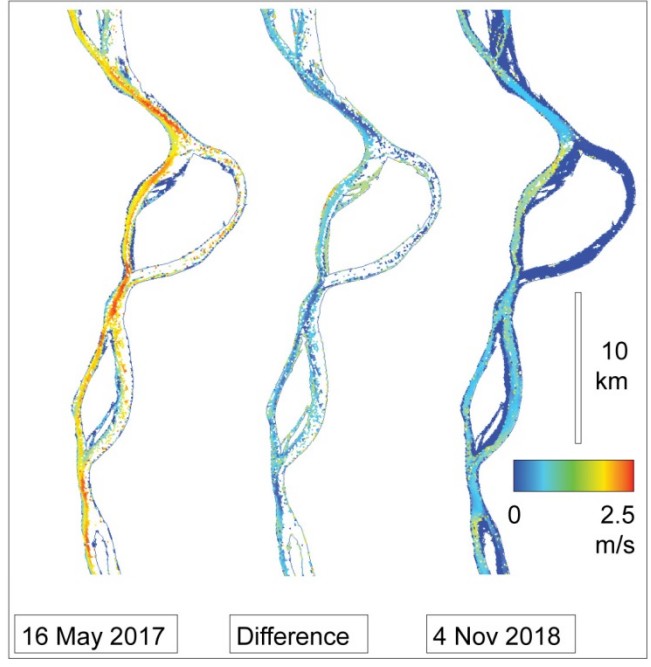

[1 column wide]

**Figure 9: Detail of water surface velocities shown in Fig. 8. For more information see caption of Fig. 8.**

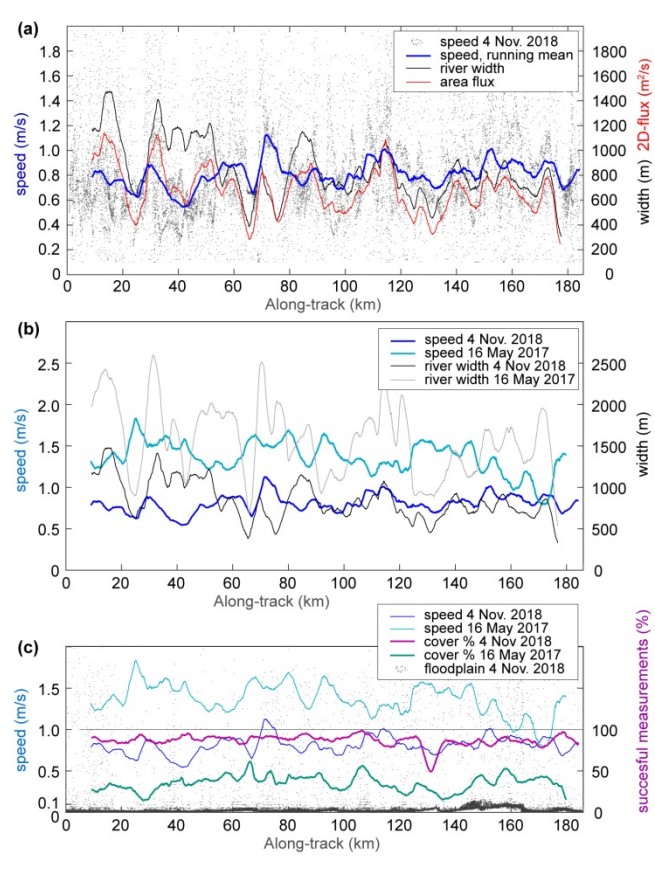

[1 column wide]

**Figure 10: Longitudinal profile of speeds and river widths derived from near-simultaneous Planet cubesat images. (a) Measurements of 4 November 2018. Small dots: individual speed measurements; blue line: 4 km running mean of individual measurements; black line: river width from velocities (running mean); red line: surface area flux as product of cross-sectional average speed and river width (running mean). (b) Running means of surface speeds and river width for 4 November 2018 (dark blue and black, respectively) and 16 May 2017 (light blue and grey, respectively). (c) Indicators of result quality. Small dots are speeds on stable ground for 4 November 2018, i.e. outside of the river. The green and turquois lines are the percentage of successful measurements (i.e. measurements passing the correlation coefficient threshold) compared to the complete river mask. Blue and light blue lines in panel (c) are the speeds as in panel (b).**

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
