# Peer review of "River ice and water velocities using the Planet optical cubesat constellation"

_Hydrology and Earth System Sciences, 2019_

## Referee Comment (RC1) · Anonymous Referee #1 · 1 May 2019

The manuscript presents very interesting results of estimation of the velocity of the ice floes on the Yukon and the Amur rivers for ice-break and ice set-up periods. The authors state that the ice velocity is retrieved with unprecedented accuracy of +- 0.01 m/s. They provide detailed and very valuable figures of across channel velocity distribution along 60 km (the Amur R.) and 200 km (the Yukon R.) reaches. For the Yukon River, the authors calculate the average velocity and measure river width along 180 km river reach and provide an estimation of the surface flux. The manuscript contains a section dedicated to the errors estimation and short discussion on difficulties of the Planet cubesat velocities retrieval and potential application of the constellation.

The manuscript provides very valuable snapshot on the river hydraulics for such a long river reaches, which cannot be measured or evaluated otherwise.

[Figure]

The manuscript is suitable for a publication on HESS. However, it needs a significant improvement.

1. The section Data and Methods needs an amelioration. More detailed (and separate) information on data used will ameliorate the reading. It seems that the authors, in addition to main Planet images dataset, use the Landsat images for river mask. However, they do not describe them in the Data section. In the section 4.2. the comparison with the ASTER derived results is made. Are these results new or already published? As it follows from the text, the only methodology is published. If the results are new, please, add their description into the Data and provide short paragraph of the method applied.

Moreover, the method section on 2/3 consists of the text cited from previous publication. I have never seen it before in journals of natural science domain and recommend rewrite this section.

2. Calculation of the mean velocity and river width is the most interesting for potential applications part of the manuscript. However, the manuscript is lack of details on the method of calculation of these parameters. How is the multi-brunch geomorphology handled in this estimation? How does the variable floe density across the river affect the estimations of both parameters? How is the floe-free areas considered? What is the accuracy of the width calculation from the ice velocity vectors considering previous issues?

3. The main accent in the manuscript is done on the Yukon River, while the Amur River is treated by side. Please, explain what the reason was. I would like to see the same details for the second river with the plot of the mean velocity and the width. As well, it will be interesting to compare in the Discussion the similar events ( freezing) on these two rivers.

4. For the Yukon River, the fig.7 presents the fields of velocity difference. What is the massage that we could retain from the difference plots? Please, explain it in the text.

[Figure]

5. Paragraph 20 on the page 16 (Discussion) repeats very interesting finding of the periodicity in spatial distribution of the velocity peak along the river, presented in paragraph 10 of the page 11. The manuscript will gain if the authors add more explanation and discussion on this phenomenon. Moreover, overall impression that the article is really lack of general Discussion and of comparison with other studies.

Other comments. line 20 page 5. "Over the limited width of rivers, water..." Please, simplify the sentence. Fig. 5. low panel. Please, explain in the sec 4.1. the noise on the islands and banks, or plot the river mask for clarity.

Figure 6. This is very interesting figure demonstrating the directions of the flow. The caption tells us that presented velocities are after thresholding of the correlation coefficient. Please, give more details. The arrows are small. If the directions can be guessed, the length (== to velocity is invisible). Please, colour the arrows.

line 25-31 page 6. Check the English.

line 1 page 8 "ice velocities" RETRIEVED "from near-simultaneous Planet"...

line 11 page 8 "The images used.....to current sensors" ... Please, explain this sentence. What does it mean?

line 12 page 10. One Landsat image of 16 Sept 2013 was used to create the mask of the Yukon River. This mask is created using blue/TIR band ration. Please, explain the choice of the bands used or give a reference on work, where the performance of this ratio was investigated.

line 21 page 10. Please, provide the standard deviation for mean discharge value at 4 November.

lines 25-31 page 10. This paragraph is rather subject for discussion section.

line 29 page 16. ICESat is not widely used for monitoring the water height in rivers as its repeat cycle is of 91 days. I would cite recently launched Sentinel -3 missions instead

of ICESat2. If the authors prefer to keep ICESat2, this will need a comprehensive discussion about potential application.

If all these questions will be addressed, I will recommend this manuscript for publication.

---

## Referee Comment (RC2) · Anonymous Referee #2 · 17 May 2019

**General comments**

The focus of the research article is the exploitation of PlanetScope constellation satellite imagery to estimate high latitude river velocities through ice floe mapping during formation and break-up periods. The authors creatively exploit an unplanned advantage provided by satellite path overlap to assemble imagery with sufficient spatial coincidence and slight temporal separation to allow velocimetry to be conducted. The potential use of PlanetScope data for this purpose is important to report. However, from methodological and interpretive standpoints, this largely reads like a rewrite of the 2011 article by the first author. The lack of methodological details, literature review and more rigorous uncertainty assessment make this read more like a technical note than a research paper. At the same time, the length required to provide pertinent details of the

constellation, which is largely a quoted excerpt from another previous work, and factors to be considered when using this technique, specifically likely sources of error, create the length associated with a research article. Caveats regarding use and sources of error are provided in a complete and succinct manner, that is much appreciated. Either the article should be shortened to technical note length by condensing much of the quoted material or revisions should be completed to make this a more useful and therefore impactful research article.

Specific comments

There have been advances in other related application areas (e.g., sea ice monitoring) that should be considered and cited here. Some recent work also cites the 2011 work of the authors – which focused on the analytical approach employed, as opposed to the input data utilized. If this is to be a research article, additional consideration of the correlation technique should be provided.

Physical interpretations of observed velocities, while logical and illustrated by the figures provided, are still rather general in nature. That is, no specific uncertainty assessment is performed. Only qualitative judgement is possible. On the one hand, the method can provide insights regarding the timing, relative magnitude, and morphological information as illustrated – so what is provided has merit. None-the-less, it is important that the procedure one would use to conduct a more rigorous uncertainty assessment be at least outlined. Even reporting the specific challenges to conducting such an analysis so would help move the science forward.

Simply put: what would be needed to convert the velocities shown to a discharge value that might be compared in more quantitative manner to recorded (or in some cases estimated) discharges? For example, in Large Scale Particle Imaging Velocimetry (LSPIV) a relationship between surface and average cross-sectional velocity (i.e., what is used in discharge estimation) is assumed (and sometimes based on calibration). There is mention of friction effects in the 2011 article, but none here. Would the authors have

suggestions regarding an appropriate approach in the case of ice floe tracking?

I believe some further discussion of data coverage by this technique is also warranted. For example, is the Yukon river study area the closest possible to the Pilot Station gauge site or have cloud cover issues prevented selection of scenes in closer proximity? This is not meant as a criticism of the work or method, only as a request to help the reader understand the potential utility of the method.

Especially as you make mention of Sentinel and Landsat satellites for potential use in this application, what are the average sizes of ice patches (or the scale lengths of features in tropical waters) necessary for them to be actually "tracked" (correlated)? I expect this has been covered by the authors in previous manuscripts, but deserves explicit mention here.

A few more comments that are more than typographical or minor grammatical ones, provided in the order in which they arise in the manuscript (as opposed to priority):

Page 2, Line 13: What constitutes "small reaches"? Please indicate the length of reaches used in the studies mentioned as the reader can't rely on figures for more specific information.

Page 7, Line 6: Please clarify what is meant by 'juxtaposed'. At first read, it is easy to presume this relates to processes discussed later in the manuscript. Do you mean that individual ice pieces are NOT colliding and landing on top of one another or twirling in a circular fashion? I find this sentence confusing. Please revise it (add several more sentences if necessary) to clarify what you mean as I suspect the point you are trying to convey is important.

Page 7, Line 9 It seems that the lowest velocities are also at the lowest elevation end of the study reach. I assume the focus is on velocity and not geography in this case. If that is correct, change "close the lower end of the river reach" to "close to the lower end of velocities for the river reach".

Page 7, line 13: What is meant by "strong and little sensitive contrast"?

Regarding figures: Figure 8 requires a legend (even though one is provided in figure 7). Figure 9 should be a little larger if possible.

Technical corrections

Page 2, Line 11 change to read '. . .ALOS PRISM sensors. Agile stereo is. . . .'

Page 2, Line 20 change 'prevent from applying the method' to ' prevent application of the method'

Page 2, Line 24 change 'second' to 'secondary'

Page 2, Line 27 change 'offers thus' to 'thus offers'

Page 2. Line 32 change 'shortly' to 'briefly'

Page 3. I don't believe it necessary to make the statement provided in parentheses or place the large sections of text in quotes. You wrote this text originally. By citing the source and providing the brief statement regarding update and specification (although I'm not sure what is meant by the latter), you can remove the quotes.

Page 4, Line 5 remove period and right parentheses between citations.

Page 5 Remove double quotation marks.

Page 5, Line 9 change "is" to "are"

Page 5, Line 18 remover "an"

Page 5, Line 21 remove 'strictly'

Page 6, Line 15 should read 'smaller than 0.7'

Page 6, Line 20 change 'estimate' to 'estimating'

Page 7, Line 6 should read: ...velocities. Ice floes directly...

Page 7, Line 7 The text on this line is confusing. Please revise, paying attention to specific comments above.

Page 10, Line 3 change "choose" to "chose"

Page 15, Line 2 change "necessary completely eliminated" to "necessarily eliminated"

Page 15, Line 5 remove comma following 'registration'

Page 15, Line 6 remove first 'actual'

Page 16, Line 4 change indicator to indicators

Page 16, Line 13 change "seems not untypical" to "seems typical'

Page 16, Line 23 change 'A major purpose of satellite observations of rivers are attempts to estimate discharge in order to spatially...' to "A major purpose of satellite-based river observations is to estimate discharge in order to spatially..."

Page 16, Line 25 remove 'validation'

Page 16, Line 30 change 'missions, and' to 'missions. And,'

Page 16, Line 30 change "actual river surface parameters" to "river width."

Page 17, Line 3 change 'and better understanding of' to 'as well as provide better understanding of'

Kaab and Leprence 2014 citation seems incomplete.

———————————————

---

## Author Comment (AC1) · 7 Jun 2019

**Hydrology and Earth System Sciences**       **hess-2019-62**

**River ice and water velocities using the Planet optical cubesat constellation**

**Andreas Kääb, Bas Altena and Joseph Mascaro**

**General response**

We would like to thank the two referees for their constructive and detailed reviews that will certainly help to improve the paper!

We agree with most of the comments made (as detailed below), and envisage no problems to modify our manuscript accordingly. In summary, we will expand method descriptions and accuracy assessment, and some data analyses/interpretations. We will add more literature about previous studies on water flow measurement/ice floe tracking. We prefer to not dig too deep into (subjectively?) selected applications of our method, as the focus of our paper is the demonstration and assessment of a measurement method.

We hope to address the below referee comments in a satisfactory way. Referee comments are in *italic*, and our response in normal font.

**Response to individual referees**

Referee #1

*The manuscript presents very interesting results of estimation of the velocity of the ice floes on the Yukon and the Amur rivers for ice-break and ice set-up periods. The authors state that the ice velocity is retrieved with unprecedented accuracy of +- 0.01 m/s. They provide detailed and very valuable figures of across channel velocity distribution along 60 km (the Amur R.) and 200 km (the Yukon R.) reaches. For the Yukon River, the authors calculate the average velocity and measure river width along 180 km river reach and provide an estimation of the surface flux. The manuscript contains a section dedicated to the errors estimation and short discussion on difficulties of the Planet cubesat velocities retrieval and potential application of the constellation.*
*The manuscript provides very valuable snapshot on the river hydraulics for such a long river reaches, which cannot be measured or evaluated otherwise.*

*The manuscript is suitable for a publication on HESS. However, it needs a significant improvement.*

Thanks a lot for this positive overall judgement of our work.

*1. The section Data and Methods needs an amelioration. More detailed (and separate) information on data used will ameliorate the reading. It seems that the authors, in addition to main Planet images dataset, use the Landsat images for river mask. However, they do not describe them in the Data section.*

We will be happy to add a paragraph on the Landsat data used, no problem. We have not done that so far as we thought that Landsat data are so established that they don't require much description.

*In the section 4.2. the comparison with*
*the ASTER derived results is made. Are these results new or already published? As it*
*follows from the text, the only methodology is published. If the results are new, please,*
*add their description into the Data and provide short paragraph of the method applied.*

Correct, the ASTER results are not published, only the method. We will be happy to add text on the data and method. The purpose of including the ASTER data and results was also to demonstrate the increased potential from the higher-resolution Planet cubesat data. We will make that clearer.

*Moreover, the method section on 2/3 consists of the text cited from previous publication.*
*I have never seen it before in journals of natural science domain and recommend rewrite*
*this section.*

This problem arose from our opinion that rewriting a technical description of an instrument (here: satellites) from an earlier paper by the same authors does not make much sense. We acknowledge the confusion, though, and will rewrite the according sections. (See also referee #2)

2. *Calculation of the mean velocity and river width is the most interesting for potential*
*applications part of the manuscript. However, the manuscript is lack of details on the*
*method of calculation of these parameters. How is the multi-brunch geomorphology*
*handled in this estimation? How does the variable floe density across the river affect*
*the estimations of both parameters? How is the floe-free areas considered? What is*
*the accuracy of the width calculation from the ice velocity vectors considering previous*
*issues?*

Thanks for this perspective! We have considered the mean velocity only as one potential application of many. The purpose of our study is to demonstrate a methodology, but leave further exploitation, and the judgment of which applications are most useful to river specialists (which we are not). New, we will give more details on the method of calculation of mean velocities and its performance. We will, however, prefer to not make this application the dominant one of our method for the reason given above. In short, multi-branch morphology can degrade the results, if not split up. Floe density variations (including floe-free areas) impact on the results, but it is possible to smooth and flag such variations in the results. This problem is much reduced under freeze-up conditions, which we are now able to capture more systematic. We will elaborate on width accuracy but think our method is not optimal and very limited to estimate river width. There are better ones, cf. Allen and Pavelsky (2018).

3. *The main accent in the manuscript is done on the Yukon River, while the Amur River*
*is treated by side. Please, explain what the reason was. I would like to see the same*
*details for the second river with the plot of the mean velocity and the width. As well, it*
*will be interesting to compare in the Discussion the similar events ( freezing) on these*
*two rivers.*

We will be happy to expand accordingly on the Amur River results, no problem. As the Amur River reach studied is "only" 60 km, the statistical significance of the results is reduced compared to the 200 km of Yukon River. A further reason for keeping the Amur River results short was that we wanted to build the paper by first demonstrating the raw method, and then performing more detailed analyses, and to use different rivers for that to include some geographic spread. Further, the Amur River reach studied is quite multi-branch, which will impact on the results (see above). We will elaborate also on these issues.

*4. For the Yukon River, the fig.7 presents the fields of velocity difference. What is the massage that we could retain from the difference plots? Please, explain it in the text.*

We will be happy to elaborate more on that. In short, the purpose of Fig 7 includes: the simple visualization of the raw results; different density of measurements; spatial variations in speed changes over time.

*5. Paragraph 20 on the page 16 (Discussion) repeats very interesting finding of the periodicity in spatial distribution of the velocity peak along the river, presented in paragraph 10 of the page 11. The manuscript will gain if the authors add more explanation and discussion on this phenomenon. Moreover, overall impression that the article is really lack of general Discussion and of comparison with other studies.*

We will in general try to expand the discussion and comparison with other studies, but also try to make clearer that specific hydrologic/hydraulic/geomorphological findings are not the purpose of the paper as we are not sure which applications of our method are most useful. Discussion with experts didn't give us a clear answer about a most promising application so far (cf. also reviewer #2 who focusses more on discharge; others seem most interested in the physical impact of ice floes on infrastructure, for instance, or validation of hydraulic models).

*Other comments.*

We will implement all below detail comments.

*line 20 page 5. "Over the limited width of rivers, water..." Please, simplify the sentence.*

*Fig. 5. low panel. Please, explain in the sec 4.1. the noise on the islands and banks, or plot the river mask for clarity.*

*Figure 6. This is very interesting figure demonstrating the directions of the flow. The caption tells us that presented velocities are after thresholding of the correlation coefficient. Please, give more details. The arrows are small. If the directions can be guessed, the length (== to velocity is invisible). Please, colour the arrows.*

*line 25-31 page 6. Check the English.*

*line 1 page 8 "ice velocities" RETRIEVED "from near-simultaneous Planet"...*

*line 11 page 8 "The images used.....to current sensors" ... Please, explain this sentence. What does it mean?*

*line 12 page 10. One Landsat image of 16 Sept 2013 was used to create the mask of the Yukon River. This mask is created using blue/TIR band ration. Please, explain the*

*choice of the bands used or give a reference on work, where the performance of this ratio was investigated.*

*line 21 page 10. Please, provide the standard deviation for mean discharge value at 4 November.*

*lines 25-31 page 10. This paragraph is rather subject for discussion section.*

*line 29 page 16. ICESat is not widely used for monitoring the water height in rivers as its repeat cycle is of 91 days. I would cite recently launched Sentinel -3 missions instead of ICESat2. If the authors prefer to keep ICESat2, this will need a comprehensive discussion about potential application.*

*If all these questions will be addressed, I will recommend this manuscript for publication.*
* * *
Referee #2

*General comments*
*The focus of the research article is the exploitation of PlanetScope constellation satellite imagery to estimate high latitude river velocities through ice floe mapping during formation and break-up periods. The authors creatively exploit an unplanned advantage provided by satellite path overlap to assemble imagery with sufficient spatial coincidence and slight temporal separation to allow velocimetry to be conducted. The potential use of PlanetScope data for this purpose is important to report. However, from methodological and interpretive standpoints, this largely reads like a rewrite of the 2011 article by the first author. The lack of methodological details, literature review and more rigorous uncertainty assessment make this read more like a technical note than a research paper. At the same time, the length required to provide pertinent details of the constellation, which is largely a quoted excerpt from another previous work, and factors to be considered when using this technique, specifically likely sources of error, create the length associated with a research article. Caveats regarding use and sources of error are provided in a complete and succinct manner, that is much appreciated. Either the article should be shortened to technical note length by condensing much of the quoted material or revisions should be completed to make this a more useful and therefore impactful research article.*

Thanks for this judgement, which is on overall consistent with the comments by referee #1. As outlined above for referee #1, we will expand methodological details and uncertainty assessment (see referee #1 comments 1 and 2). We will also expand the literature review, assuming the referee means technical studies about measurement of river velocities from space, as this is the main focus of our study. We will rewrite the description of the Planet cubesat constellation (referee #1 comment 1). Given the below comments and those of referee #1 we prefer to improve the manuscript towards a research article, as we else would not be able to respond adequately to all comments.

*Specific comments*

*There have been advances in other related application areas (e.g., sea ice monitoring)*

*that should be considered and cited here. Some recent work also cites the 2011 work of the authors – which focused on the analytical approach employed, as opposed to the input data utilized. If this is to be a research article, additional consideration of the correlation technique should be provided.*

We will be happy to expand on literature on ice floe motion from space (see above general response), and on the correlation technique used (remark: standard normalized cross-correlation, nothing special).

*Physical interpretations of observed velocities, while logical and illustrated by the figures provided, are still rather general in nature. That is, no specific uncertainty assessment is performed. Only qualitative judgement is possible. On the one hand, the method can provide insights regarding the timing, relative magnitude, and morphological information as illustrated – so what is provided has merit. None-the-less, it is important that the procedure one would use to conduct a more rigorous uncertainty assessment be at least outlined. Even reporting the specific challenges to conducting such an analysis so would help move the science forward.*

We will elaborate more on uncertainty assessment and related challenges (see also referee #1 comment 2).

*Simply put: what would be needed to convert the velocities shown to a discharge value that might be compared in more quantitative manner to recorded (or in some cases estimated) discharges? For example, in Large Scale Particle Imaging Velocimetry (LSPIV) a relationship between surface and average cross-sectional velocity (i.e., what is used in discharge estimation) is assumed (and sometimes based on calibration). There is mention of friction effects in the 2011 article, but none here. Would the authors have suggestions regarding an appropriate approach in the case of ice floe tracking?*

We agree with the referee that discharge estimates from our measurements could be a potentially interesting application, and will be happy to elaborate a bit more on that. We have demonstrated the principal feasibility of ice floe tracking for discharge estimates already in Beltaos and Kääb (2014). We hesitate, however, to focus in the present paper too much on that one application as the focus of our study is not directed to a selected specific application. There are other potential applications (see referee #1 comment 5; e.g., river morphology, engineering, hydraulic modelling) and we prefer our manuscript to be open in that respect.

*I believe some further discussion of data coverage by this technique is also warranted. For example, is the Yukon river study area the closest possible to the Pilot Station gauge site or have cloud cover issues prevented selection of scenes in closer proximity? This is not meant as a criticism of the work or method, only as a request to help the reader understand the potential utility of the method.*

We will elaborate more on actual coverage by useful data. We presented actually some similar data near Pilot Station at AGU2018, but found that a sound comparison of our measurements to discharge measurements/estimates requires more focus on hydraulic relations than reasonable within the intended focus of this manuscript, and that waiting a bit longer to collect more repeat data would further strengthen the analysis. Certainly, there are other reaches that are even better suited for such work (with available discharge, bathymetry, etc.; e.g., Beltaos and Kääb, 2014).

*Especially as you make mention of Sentinel and Landsat satellites for potential use*

*in this application, what are the average sizes of ice patches (or the scale lengths of features in tropical waters) necessary for them to be actually "tracked" (correlated)? I expect this has been covered by the authors in previous manuscripts, but deserves explicit mention here.*

We will elaborate more on this, no problem.

*A few more comments that are more than typographical or minor grammatical ones, provided in the order in which they arise in the manuscript (as opposed to priority):*

*Page 2, Line 13: What constitutes "small reaches"? Please indicate the length of reaches used in the studies mentioned as the reader can't rely on figures for more specific information.*

We will specify (a few tens of km). We meant "short" with respect to 600 km mentioned in the following sentence.

*Page 7, Line 6: Please clarify what is meant by 'juxtaposed'. At first read, it is easy to presume this relates to processes discussed later in the manuscript. Do you mean that individual ice pieces are NOT colliding and landing on top of one another or twirling in a circular fashion? I find this sentence confusing. Please revise it (add several more sentences if necessary) to clarify what you mean as I suspect the point you are trying to convey is important.*

We mean 'not colliding', and will clarify and describe in more detail.

*Page 7, Line 9 It seems that the lowest velocities are also at the lowest elevation end of the study reach. I assume the focus is on velocity and not geography in this case. If that is correct, change "close the lower end of the river reach" to "close to the lower end of velocities for the river reach".*

We mean the geographic location where the maximum speeds are found (=lowest elevation of the study reach). We will clarify.

*Page 7, line 13: What is meant by "strong and little sensitive contrast"?*

We will clarify.

*Regarding figures: Figure 8 requires a legend (even though one is provided in figure 7). Figure 9 should be a little larger if possible.*

We will change.

*Technical corrections*

We will implement all below detail comments.

*Page 2, Line 11 change to read ': : :ALOS PRISM sensors. Agile stereo is: : :.'*
*Page 2, Line 20 change 'prevent from applying the method' to ' prevent application of the method'*
*Page 2, Line 24 change 'second' to 'secondary'*

*Page 2, Line 27 change 'offers thus' to 'thus offers'*
*Page 2. Line 32 change 'shortly' to 'briefly'*
*Page 3. I don't believe it necessary to make the statement provided in parentheses or place the large sections of text in quotes. You wrote this text originally. By citing the source and providing the brief statement regarding update and specification (although I'm not sure what is meant by the latter), you can remove the quotes.*

It turned out that editors, referees, and authors of this manuscript have all different opinions about how to deal with a technical description of an instrument by the same authors from an earlier publication. To avoid this confusion we will rewrite the text of concern. (See also referee #1 comment 1).

*Page 4, Line 5 remove period and right parentheses between citations.*
*Page 5 Remove double quotation marks.*
*Page 5, Line 9 change "is" to "are"*
*Page 5, Line 18 remover "an"*
*Page 5, Line 21 remove 'strictly'*
*Page 6, Line 15 should read 'smaller than 0.7'*
*Page 6, Line 20 change 'estimate' to 'estimating'*
*Page 7, Line 6 should read: ...velocities. Ice floes directly...*
*Page 7, Line 7 The text on this line is confusing. Please revise, paying attention to specific comments above.*
*Page 10, Line 3 change "choose" to "chose"*
*Page 15, Line 2 change "necessary completely eliminated" to "necessarily eliminated"*
*Page 15, Line 5 remove comma following 'registration'*
*Page 15, Line 6 remove first 'actual'*
*Page 16, Line 4 change indicator to indicators*
*Page 16, Line 13 change "seems not untypical" to "seems typical"*
*Page 16, Line 23 change 'A major purpose of satellite observations of rivers are attempts to estimate discharge in order to spatially: : :' to "A major purpose of satellitebased river observations is to estimate discharge in order to spatially: : :"*
*Page 16, Line 25 remove 'validation'*
*Page 16, Line 30 change 'missions, and' to 'missions. And,'*
*Page 16, Line 30 change "actual river surface parameters" to "river width."*
*Page 17, Line 3 change 'and better understanding of' to 'as well as provide better understanding of'*
*Kaab and Leprence 2014 citation seems incomplete.*

---

## Editor Comment (EC1) · Bettina Schaefli (Editor) · 18 Jun 2019

Both reviewers agree that the manuscript is suitable for HESS but that it lacks methodological and literature details for a full research paper. I invite the authors to submit a revised paper that addresses these shortcomings (along the lines in the public discussion).
* * *

---

## Author Response (AR1)

Hydrology and Earth System Sciences  hess-2019-62
River ice and water velocities using the Planet optical cubesat constellation
Andreas Kääb, Bas Altena and Joseph Mascaro

**Response to referees and description of revisions**

**General response**

We would like to thank the two referees for their constructive and detailed reviews that certainly helped to improve the paper!

We agree with most of the comments made (as detailed below), and modified our manuscript accordingly. In summary, we expanded method descriptions and accuracy assessment, and some data analyses/interpretations. We added more literature about previous studies on water flow measurement/ice floe tracking. We preferred to dig not too deep into selected applications of our method, as the focus of our paper is the demonstration and assessment of a measurement method.

We hope to have addressed the below referee comments in a satisfactory way. Referee comments are in *italic*, and our response/revisions in normal font. A manuscript version with all changes made in red is attached.

**Response to individual referees**

Referee #1

*The manuscript presents very interesting results of estimation of the velocity of the ice floes on the Yukon and the Amur rivers for ice-break and ice set-up periods. The authors state that the ice velocity is retrieved with unprecedented accuracy of +- 0.01 m/s. They provide detailed and very valuable figures of across channel velocity distribution along 60 km (the Amur R.) and 200 km (the Yukon R.) reaches. For the Yukon River, the authors calculate the average velocity and measure river width along 180 km river reach and provide an estimation of the surface flux. The manuscript contains a section dedicated to the errors estimation and short discussion on difficulties of the Planet cubesat velocities retrieval and potential application of the constellation.*
*The manuscript provides very valuable snapshot on the river hydraulics for such a long river reaches, which cannot be measured or evaluated otherwise.*

*The manuscript is suitable for a publication on HESS. However, it needs a significant improvement.*

Thanks a lot for this positive overall judgement of our work.

*1. The section Data and Methods needs an amelioration. More detailed (and separate) information on data used will ameliorate the reading. It seems that the authors, in addition*

*to main Planet images dataset, use the Landsat images for river mask. However, they do not describe them in the Data section.*

We added a paragraph on the Landsat (and ASTER) data used.

*In the section 4.2. the comparison with the ASTER derived results is made. Are these results new or already published? As it follows from the text, the only methodology is published. If the results are new, please, add their description into the Data and provide short paragraph of the method applied.*

Correct, the ASTER results are not published, only the method. We added text on the data and method. The purpose of including the ASTER data and results was also to demonstrate the increased potential from the higher-resolution Planet cubesat data. We made that clearer.

*Moreover, the method section on 2/3 consists of the text cited from previous publication. I have never seen it before in journals of natural science domain and recommend rewrite this section.*

This problem arose from our opinion that rewriting a technical description of an instrument (here: particular satellites) from an earlier paper by the same authors does not make much sense. We acknowledge the confusion, though, and rewrote the according sections. (See also referee #2)

*2. Calculation of the mean velocity and river width is the most interesting for potential applications part of the manuscript. However, the manuscript is lack of details on the method of calculation of these parameters. How is the multi-brunch geomorphology handled in this estimation? How does the variable floe density across the river affect the estimations of both parameters? How is the floe-free areas considered? What is the accuracy of the width calculation from the ice velocity vectors considering previous issues?*

Thanks for this perspective! We have considered the mean velocity only as one potential application of many. The purpose of our study is to demonstrate a methodology, but leave further exploitation, and the judgment of which applications are most useful to river specialists. New, we give more details on the method of calculation of mean velocities and its performance, and discuss modifications. We also added a figure that demonstrates the performance (Fig 10c). We preferred, however, to not make this application the primary one of our method for the reason given above.

*3. The main accent in the manuscript is done on the Yukon River, while the Amur River is treated by side. Please, explain what the reason was. I would like to see the same details for the second river with the plot of the mean velocity and the width. As well, it will be interesting to compare in the Discussion the similar events ( freezing) on these two rivers.*

We added a longitudinal profile of mean velocities for Amur River and some description of it (new Fig. 7). As the Amur River reach studied is "only" 60 km, the statistical significance of the results is reduced compared to the 200 km of Yukon River. A further reason for keeping the Amur River results short was that we wanted to build the paper by first demonstrating the raw method, and then performing more detailed analyses, and to use different rivers for that to include some geographic spread. Further, the Amur River reach studied is quite multi-branch, which impacts on the results (see above). We mention now also this issue.

*4. For the Yukon River, the fig.7 presents the fields of velocity difference. What is the massage that we could retain from the difference plots? Please, explain it in the text.*

We elaborated more on that. In short, the purpose of the figure includes: the simple visualization of the raw results; different density of measurements; spatial variations in speed changes over time.

*5. Paragraph 20 on the page 16 (Discussion) repeats very interesting finding of the periodicity in spatial distribution of the velocity peak along the river, presented in paragraph 10 of the page 11. The manuscript will gain if the authors add more explanation and discussion on this phenomenon. Moreover, overall impression that the article is really lack of general Discussion and of comparison with other studies.*

We tried to expand the discussion and comparison with other studies, but also tried to make clearer that specific hydrologic/hydraulic/geomorphological findings are not the purpose of the paper as we are not sure which applications of our method are most useful. Discussion with experts did not give us a clear answer about a most promising application so far (cf. also reviewer #2 who focusses more on discharge; others seem most interested in the physical impact of ice floes on infrastructure, for instance, or validation of hydraulic models).

*Other comments.*

We implemented all below detail comments.

*line 20 page 5. "Over the limited width of rivers, water..." Please, simplify the sentence.* Done

*Fig. 5. low panel. Please, explain in the sec 4.1. the noise on the islands and banks, or plot the river mask for clarity.* Done

*Figure 6. This is very interesting figure demonstrating the directions of the flow. The caption tells us that presented velocities are after thresholding of the correlation coefficient. Please, give more details. The arrows are small. If the directions can be guessed, the length (== to velocity is invisible). Please, colour the arrows.* Done. After graphical tests we preferred a combination of color-coded speed with vectors superimposed.

*line 25-31 page 6. Check the English.* Done

*line 1 page 8 "ice velocities" RETRIEVED "from near-simultaneous Planet"...* Done

*line 11 page 8 "The images used.....to current sensors" ... Please, explain this sentence. What does it mean?* Done

*line 12 page 10. One Landsat image of 16 Sept 2013 was used to create the mask of the Yukon River. This mask is created using blue/TIR band ration. Please, explain the choice of the bands used or give a reference on work, where the performance of this ratio was investigated.* Done

*line 21 page 10. Please, provide the standard deviation for mean discharge value at 4 November.* Meanwhile the 4 Nov 2018 discharge value is available and is now given.

*lines 25-31 page 10. This paragraph is rather subject for discussion section.* Moved

*line 29 page 16. ICESat is not widely used for monitoring the water height in rivers as its repeat cycle is of 91 days. I would cite recently launched Sentinel -3 missions instead of ICESat2. If the authors prefer to keep ICESat2, this will need a comprehensive discussion about potential application.* We included Sentinel-3. We also kept ICESat-2 but added a reference to its potential water applications.

*If all these questions will be addressed, I will recommend this manuscript for publication.*

Referee #2

*General comments*
*The focus of the research article is the exploitation of PlanetScope constellation satellite imagery to estimate high latitude river velocities through ice floe mapping during formation and break-up periods. The authors creatively exploit an unplanned advantage provided by satellite path overlap to assemble imagery with sufficient spatial coincidence and slight temporal separation to allow velocimetry to be conducted. The potential use of PlanetScope data for this purpose is important to report. However, from methodological and interpretive standpoints, this largely reads like a rewrite of the 2011 article by the first author. The lack of methodological details, literature review and more rigorous uncertainty assessment make this read more like a technical note than a research paper. At the same time, the length required to provide pertinent details of the constellation, which is largely a quoted excerpt from another previous work, and factors to be considered when using this technique, specifically likely sources of error, create the length associated with a research article. Caveats regarding use and sources of error are provided in a complete and succinct manner, that is much appreciated. Either the article should be shortened to technical note length by condensing much of the quoted material or revisions should be completed to make this a more useful and therefore impactful research article.*

Thanks for this judgement, which is on overall consistent with the comments by referee #1. As outlined above for referee #1, we expanded methodological details and uncertainty assessments (see referee #1 comments 1 and 2). We also expanded the literature review, assuming the referee means technical studies about measurement of river velocities from space, as this is the main focus of our study. We rewrote the description of the Planet cubesat constellation (referee #1 comment 1). Given the below comments and those of referee #1 we prefer to improve the manuscript towards a research article, as we else would not be able to respond adequately to all comments.

*Specific comments*

*There have been advances in other related application areas (e.g., sea ice monitoring) that should be considered and cited here. Some recent work also cites the 2011 work of the authors – which focused on the analytical approach employed, as opposed to the input data utilized. If this is to be a research article, additional consideration of the correlation technique should be provided.*

We expanded on literature (mostly in the discussion section; see above general response), and on the correlation technique used (remark: standard normalized cross-correlation, nothing special).

*Physical interpretations of observed velocities, while logical and illustrated by the figures provided, are still rather general in nature. That is, no specific uncertainty assessment is performed. Only qualitative judgement is possible. On the one hand, the method can provide insights regarding the timing, relative magnitude, and morphological information as illustrated – so what is provided has merit. None-the-less, it is important that the procedure one would use to conduct a more rigorous uncertainty assessment be at least outlined. Even reporting the specific challenges to conducting such an analysis so would help move the science forward.*

We elaborated more on uncertainty assessment and related challenges (see also referee #1 comment 2).

*Simply put: what would be needed to convert the velocities shown to a discharge value that might be compared in more quantitative manner to recorded (or in some cases estimated) discharges? For example, in Large Scale Particle Imaging Velocimetry (LSPIV) a relationship between surface and average cross-sectional velocity (i.e., what is used in discharge estimation) is assumed (and sometimes based on calibration). There is mention of friction effects in the 2011 article, but none here. Would the authors have suggestions regarding an appropriate approach in the case of ice floe tracking?*

We agree with the referee that discharge estimates from our measurements could be a potentially interesting application. We have demonstrated the principal feasibility of ice floe tracking for discharge estimates already in Beltaos and Kääb (2014). We hesitate however to focus in the present paper too much on that one application, as the focus of our study is not directed to a selected specific application. There are other potential applications (see referee #1 comment 5; e.g., river morphology, engineering, hydraulic modelling) and we prefer our manuscript to be open in that respect.

*I believe some further discussion of data coverage by this technique is also warranted. For example, is the Yukon river study area the closest possible to the Pilot Station gauge site or have cloud cover issues prevented selection of scenes in closer proximity? This is not meant as a criticism of the work or method, only as a request to help the reader understand the potential utility of the method.*

We elaborated more on actual coverage by useful data. We presented actually some similar data near Pilot Station at AGU2018, but found that a sound comparison of our measurements to discharge measurements/estimates requires more focus on hydraulic relations than reasonable within the intended focus of this manuscript, and that waiting a bit longer to collect more repeat data would further strengthen such analysis. Certainly, there are other reaches that are even better suited for such work (with available discharge, bathymetry, etc.; e.g., Beltaos and Kääb, 2014).

*Especially as you make mention of Sentinel and Landsat satellites for potential use in this application, what are the average sizes of ice patches (or the scale lengths of features in tropical waters) necessary for them to be actually "tracked" (correlated)? I expect this has been covered by the authors in previous manuscripts, but deserves explicit mention here.*

We elaborated more on this.

*A few more comments that are more than typographical or minor grammatical ones, provided in the order in which they arise in the manuscript (as opposed to priority):*

*Page 2, Line 13: What constitutes "small reaches"? Please indicate the length of reaches used in the studies mentioned as the reader can't rely on figures for more specific information.*

We specified (a few tens of km). We meant "short" with respect to 600 km mentioned in the following sentence.

*Page 7, Line 6: Please clarify what is meant by 'juxtaposed'. At first read, it is easy to presume this relates to processes discussed later in the manuscript. Do you mean that individual ice pieces are NOT colliding and landing on top of one another or twirling in a circular fashion? I find this sentence confusing. Please revise it (add several more sentences if necessary) to clarify what you mean as I suspect the point you are trying to convey is important.*

We mean 'not colliding'. We clarified and described in more detail.

*Page 7, Line 9 It seems that the lowest velocities are also at the lowest elevation end of the study reach. I assume the focus is on velocity and not geography in this case. If that is correct, change "close the lower end of the river reach" to "close to the lower end of velocities for the river reach".*

We mean the geographic location where the maximum speeds are found (=lowest elevation of the study reach). We clarified.

*Page 7, line 13: What is meant by "strong and little sensitive contrast"?*

Clarified to: "Despite these two complications, matches of ice floes seem robust with accuracy and reliability little affected because the bright floes offer particularly strong visual contrast against the surrounding dark water surface."

*Regarding figures: Figure 8 requires a legend (even though one is provided in figure 7). Figure 9 should be a little larger if possible.*

Changed.

*Technical* corrections

We implemented all below corrections.

*Page 2, Line 11 change to read ': : :ALOS PRISM sensors. Agile stereo is: : :.'*Done
*Page 2, Line 20 change 'prevent from applying the method' to ' prevent application of the method'* Done
*Page 2, Line 24 change 'second' to 'secondary'* Done
*Page 2, Line 27 change 'offers thus' to 'thus offers'* Done
*Page 2. Line 32 change 'shortly' to 'briefly'* Done
*Page 3. I don't believe it necessary to make the statement provided in parentheses or place the large sections of text in quotes. You wrote this text originally. By citing the source and providing the brief statement regarding update and specification (although I'm not sure what is meant by the latter), you can remove the quotes.*

It turned out that editors, referees, and authors of this manuscript have all different opinions about how to deal with a technical description of an instrument by the same authors from an earlier publication. To avoid this confusion we rewrote the text of concern. (See also referee #1 comment 1).

*Page 4, Line 5 remove period and right parentheses between citations.* Done
*Page 5 Remove double quotation marks.* Done
*Page 5, Line 9 change "is" to "are"* Done

*Page 5, Line 18 remover "an"* Done
*Page 5, Line 21 remove 'strictly'* Done
*Page 6, Line 15 should read 'smaller than 0.7'* Done
*Page 6, Line 20 change 'estimate' to 'estimating'* Done
*Page 7, Line 6 should read: ...velocities. Ice floes directly...* Done
*Page 7, Line 7 The text on this line is confusing. Please revise, paying attention to specific comments above.* Done
*Page 10, Line 3 change "choose" to "chose"* Done
*Page 15, Line 2 change "necessary completely eliminated" to "necessarily eliminated"* Done
*Page 15, Line 5 remove comma following 'registration'* Done
*Page 15, Line 6 remove first 'actual'* Done
*Page 16, Line 4 change indicator to indicators* Done
*Page 16, Line 13 change "seems not untypical" to "seems typical"* Done
*Page 16, Line 23 change 'A major purpose of satellite observations of rivers are attempts to estimate discharge in order to spatially: : :' to "A major purpose of satellitebased river observations is to estimate discharge in order to spatially: : :"* Done
*Page 16, Line 25 remove 'validation'* Done
*Page 16, Line 30 change 'missions, and' to 'missions. And,'* Done
*Page 16, Line 30 change "actual river surface parameters" to "river width."* Done
*Page 17, Line 3 change 'and better understanding of' to 'as well as provide better understanding of'* Done
*Kaab and Leprence 2014 citation seems incomplete.* Done

[revised manuscript text omitted]